# KAMBAAD: ENHANCING STATE SPACE MODELS WITH KOLMOGOROV–ARNOLD FOR TIME SERIES ANOMALY DETECTION

## ABSTRACT

Time series anomaly detection is critical in numerous practical applications, yet existing deep learning methods often fall short of real-world demands. These models fail to swiftly filter out physically implausible anomalies, insufficiently address distributional shifts, and lack a comprehensive approach that integrates both global and local perspectives for anomaly detection. Moreover, most successful models rely on channel-dependent methods that tend to treat all features at the same timestamp as a single token and then focus on finding relationships between these tokens. This approach overlooks the unique periodicities, trends, and lagged relationships between different features, leading to suboptimal performance. To address these limitations, we propose KambaAD, a model comprised of an Encoder and Reconstructor. The Encoder integrates the strengths of the Kolmogorov-Arnold Network (KAN), attention mechanism, and the Selective Structured State Space Model (MAMBA). Specifically, KAN is employed to swiftly enforce data consistency, enabling rapid detection of anomalies that violate physical laws. Attention mechanism ensures balanced processing of global information while enhancing the representation of key data characteristics. We leverage MAMBA's capability as a sequence model to capture anomalies caused by local variations. Additionally, its internal selection mechanism allows the model to effectively handle distribution shifts, ensuring robustness and adaptability in the presence of changing data distributions. Additionally, the framework incorporates a time-series-specific Reconstructor, which reduces computational complexity through patch-based operations that exploit local consistency in time series data. It also employs channel-independent linear reconstruction to prevent interference between different features. Through extensive experiments on multiple multivariate datasets, KambaAD consistently outperforms state-of-the-art models, demonstrating its superior performance in anomaly detection.

## 1 INTRODUCTION

Time series anomaly detection aims to accurately identify points or subsequences that deviate from regular patterns within continuous time series data. In the context of digital operations management, this technology is essential for tracking key performance indicators (KPIs) such as CPU utilization, memory usage, and network bandwidth, which generate large volumes of time series data (Zhu et al., 2019). By detecting anomalies such as performance bottlenecks or system failures, operations teams can swiftly mitigate issues, ensuring system resilience, scalability, and high availability while reducing downtime and failure rates (Lindemann et al., 2021). Beyond digital infrastructure, time series anomaly detection is also applied in fields such as economics, meteorology, and finance. For instance, it helps detect abnormal market fluctuations, predict extreme weather events, and identify fraudulent financial transactions (Ahmed et al., 2016; Lee et al., 2018; Hilal et al., 2022). Traditional anomaly detection methods rely on handcrafted features and statistical assumptions, offering simplicity and interpretability but struggling with scalability and adaptability in diverse or high-dimensional datasets (Teng, 2010; Munir et al., 2019).

In recent decades, deep learning (DL) methods have been widely adopted for time series anomaly detection, excelling at capturing temporal dependencies and nonlinear relationships without manual

feature engineering (Choi et al., 2021). However, DL methods face several challenges: Firstly, they often overlook the overall consistency of the data, making them prone to missing subtle or physically implausible anomalies. If these anomalies are not identified promptly, the model may confuse normal and abnormal patterns, especially when anomalies densely occur within a specific window, as it tries to minimize the overall error for that window. Second, DL methods struggle with distributional shifts over time, where patterns of normality and anomaly evolve. Anomalies during training may become normal later, while new anomalies emerge from previously normal patterns, complicating model robustness in dynamic, non-stationary data (Zeng, 2020; Wen & Keyes, 2019). Third, DL methods often fail to integrate both global and local perspectives—essential for detecting long-term deviations and short-term fluctuations, respectively (Xia et al., 2023; Albu et al., 2020). Finally, most successful models adopt channel-dependent strategies that treat features at the same timestamp as a single token, disregarding distinct periodicities, trends, and lagged relationships. This limits their ability to capture complex interactions in multivariate time series, leading to suboptimal reconstruction (Liu et al., 2022).

To address the multifaceted challenges in time series anomaly detection, we propose KambaAD, a robust framework composed of an Encoder and a Reconstructor. The Encoder employs a two-stage feature extraction process: Coarse-Grained Anomaly Filtering and Fine-Grained Pattern Recognition. In the Coarse-Grained Anomaly Filtering stage, KAN replaces traditional weight parameters with learnable univariate functions, establishing more stable functional relationships between inputs and outputs. This not only reduces the number of parameters but also leverages data consistency, enabling faster preliminary anomaly screening. By processing features from all time steps within the entire time window, KAN effectively captures latent cross-temporal correlations, avoiding issues such as lag effects, periodicity, and feature misalignment that arise from analyzing individual data points. Once KAN has addressed the more apparent anomalies, the process transitions seamlessly to the Fine-Grained Pattern Recognition stage, where the focus shifts to detecting more nuanced and infrequent anomalies. In this stage, we integrate attention mechanism and MAMBA within the representation space. Attention mechanism excels at capturing global dependencies across distant time steps, making it particularly useful for detecting long-range correlations in complex time series (Matar et al., 2023). Meanwhile, MAMBA focuses on detecting local, context-specific anomalies. As a sequential model, MAMBA iteratively updates its hidden states while leveraging positional information to capture temporal dependencies. Its dynamic adjustment mechanism adapts the state transition matrix to input changes, addressing distribution shifts. MAMBA's gating mechanism further modulates the influence of inputs on hidden states and outputs, reducing the impact of anomalous points on the reconstruction process. Additionally, MAMBA employs 1D convolutional operations to efficiently capture local dependencies between adjacent time steps, enhancing its ability to detect short-term anomalies and trend shifts. In the Reconstructor, we employ three complementary techniques: patch-based data division, channel-independent (CI) processing, and linear reconstruction. First, patch-based division allows the model to leverage local similarities, focusing on critical temporal features rather than processing the entire sequence at once, thereby reducing resource consumption (Berral et al., 2021; Scherer et al., 2021; Sabater et al., 2022). Second, the CI strategy ensures that each channel is processed independently during reconstruction, allowing the model to refine individual features without losing the global context, as inter-channel relationships have already been captured in the Encoder. Finally, linear reconstruction further controls the number of parameters, ensuring the model remains scalable and robust when handling high-dimensional data.

In summary, the integration of KAN, attention mechanism, and MAMBA in KambaAD enables effective coarse-grained anomaly filtering and fine-grained pattern recognition across temporal scales. Extensive experiments on multiple datasets show a 5% improvement in F1 score, confirming its effectiveness in time series anomaly detection.

## 2 PROBLEM DEFINITION

Consider a multivariate time series with $k$ variables over $t$ time steps. Each observation at time $t$ is represented by the vector $\mathbf{x}_t = (x_{t1}, x_{t2}, \ldots, x_{tk})$, where $x_{tk}$ denotes the value of the $k$-th variable at time $t$. The goal of anomaly detection is to determine whether $\mathbf{x}_t$ is normal or anomalous, based on a sliding window of the past $n$ observations, including the current time step $t$. This sliding window, denoted by $\mathbf{X}_t$, contains the vectors from time $t - n + 1$ to $t$. The anomaly detection function $f$

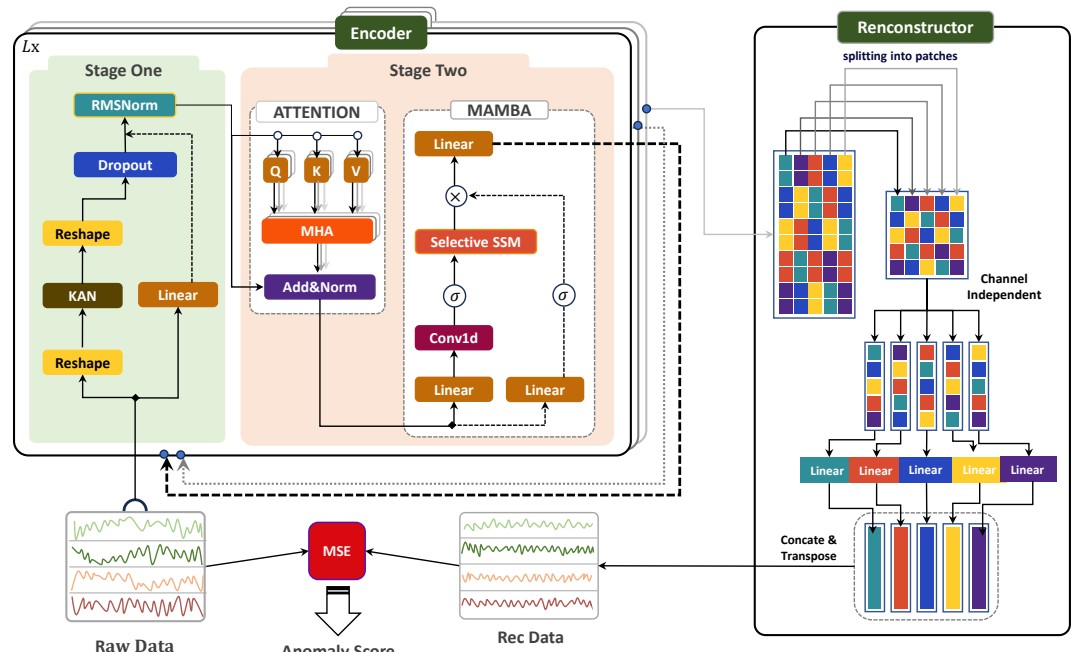

Figure 1: Architecture of KambaAD. Raw Data represents the input segmented into windows, while Rec Data refers to the reconstructed output after model processing.

maps this sliding window $\mathbf{X}_t$ to a binary label $y_t$:

$$y_t = f(\mathbf{X}_t) = \begin{cases} 1 & \text{anomaly} \\ 0 & \text{normal.} \end{cases} \tag{1}$$

## 3 METHODOLOGY

### 3.1 OVERVIEW

As shown in Figure 1, the model processes windowed data as input and outputs an anomaly score. The architecture consists of two parts: feature extraction and reconstruction. In the feature extraction phase, a hybrid encoder combining KAN, attention mechanisms, and MAMBA is used. In the reconstruction phase, the data is divided into patches and fed into channel-independent linear models for reconstruction. The reconstruction error is computed as the mean squared error (MSE) between the input and the reconstructed output.

### 3.2 ENCODER

This section elucidates the mathematical underpinnings of the KambaAD encoder component. Let the input data be denoted as $\mathbf{X} \in \mathbb{R}^{n \times k}$, where $n$ represents the number of data points and $k$ signifies the number of features. Our primary objective is to extract information-rich features capable of reconstructing the original data with high fidelity.

#### 3.2.1 STAGE ONE: COARSE-GRAINED ANOMALY FILTERING

In this study, we introduce the use of the KAN for preliminary anomaly detection in time series data. To tackle challenges such as varying periodicities, trends, and lag effects across features, we consider the entire window of features as input, allowing the model to capture relationships across different time steps. The input data is represented as a matrix $\mathbf{X} \in \mathbb{R}^{n \times k}$, where each data point $X_i = \{x_1, x_2, \ldots, x_k\}$ (for $i = 1, 2, \ldots, n$). The operations in this phase are denoted by $F_{\text{KAN}}$, where $\mathbf{X}$ is first reshaped into a vector, processed, and then reshaped back into an $n \times d$ matrix. For

details on the KAN architecture, please refer to Appendix C. After processing through $F_{\text{KAN}}$, we apply post-processing steps such as dropout (Srivastava et al., 2014), linear residual combination, and Root Mean Square Normalization (RMSNorm) (Zhang & Sennrich, 2019). This post-processing is formalized as follows:

$$\mathbf{X}_1 = \mathbf{X}\mathbf{W}_\alpha + \mathbf{b}_\alpha, \tag{2}$$

where $\mathbf{X}_1 \in \mathbb{R}^{n \times d}$ represents the linearly transformed residuals, adjusted to the universal dimension $d$. This transformation prepares the residuals for combination with the main output of KAN. Here, $\mathbf{W}_\alpha \in \mathbb{R}^{k \times d}$ and $\mathbf{b}_\alpha \in \mathbb{R}^{n \times d}$ denote the weight matrix and bias vector, respectively. The final output of the model, $\mathbf{X}_{\text{KAN}} \in \mathbb{R}^{n \times d}$, which captures anomalies in the data, is expressed as:

$$\mathbf{X}_{\text{KAN}} = \text{RMSNorm}(\text{Dropout}(F_{\text{KAN}}(\mathbf{X})) + \mathbf{X}_1). \tag{3}$$

### 3.2.2 STAGE TWO: FINE-GRAINED PATTERN RECOGNITION

In this phase, we refine anomaly detection by first applying a self-attention mechanism to provide a global perspective, enhancing time series features and highlighting anomalies. Next, the MAMBA model performs recursive updates to capture subtle temporal variations. This two-step process balances long-range dependency capture with local anomaly detection. By combining the global pattern recognition of self-attention with MAMBA's efficient modeling of local dynamics, our framework effectively detects both prominent and subtle anomalies. We first apply a multi-head attention mechanism to the time series data, transforming the input, calculating attention scores, and aggregating the results. For each attention head, the output is computed using the query ($\mathbf{Q}$), key ($\mathbf{K}$), and value ($\mathbf{V}$) matrices as follows:

$$\text{head}_i = \text{softmax}\left(\frac{\mathbf{Q}_i\mathbf{K}_i^T}{\sqrt{d_k}}\right)\mathbf{V}_i, \tag{4}$$

where the queries, keys, and values are obtained as:

$$\mathbf{Q}_i = \mathbf{X}_{\text{KAN}}\mathbf{W}_i^Q, \quad \mathbf{K}_i = \mathbf{X}_{\text{KAN}}\mathbf{W}_i^K, \quad \mathbf{V}_i = \mathbf{X}_{\text{KAN}}\mathbf{W}_i^V, \tag{5}$$

with $\mathbf{W}_i^Q$, $\mathbf{W}_i^K$, and $\mathbf{W}_i^V$ being learnable weight matrices, and $d_k$ representing the dimensionality of the key vectors. The outputs from all attention heads are concatenated and linearly transformed to form the final output $\mathbf{O}$. This output is combined with the original input $\mathbf{X}_{\text{KAN}}$ through a residual connection, followed by dropout and layer normalization:

$$\mathbf{X}_{\text{Attn}} = \text{RMSNorm}(\text{Dropout}(\mathbf{O}) + \mathbf{X}_{\text{KAN}}), \tag{6}$$

where $\mathbf{X}_{\text{Attn}} \in \mathbb{R}^{n \times d}$ represents the output from attention layer. Subsequently, the data is processed through the MAMBA module, which is specifically designed to detect anomalies by identifying subtle local variations in the input sequence. The input data, denoted as $\mathbf{X}_{\text{Attn}}$, undergoes a series of transformations, including linear projection, 1D convolution, and the SiLU (Swish) activation function. This process produces the intermediate representation $\mathbf{X_2} \in \mathbb{R}^{n \times d}$, which serves as the input to the structured state-space model (SSM) module. Mathematically, this transformation can be expressed as:

$$\mathbf{X_2} = \text{SiLU}\left(\text{Conv1D}\left(\mathbf{X}_{\text{Attn}}\mathbf{W}_\beta, h\right)\right), \tag{7}$$

where $\mathbf{W}_\beta$ is the learnable linear transformation matrix, $h$ is the 1D convolutional kernel, and $\text{SiLU}(x) = x \cdot \sigma(x)$ (with $\sigma(x)$ being the sigmoid function) is the activation function. The resulting matrix $\mathbf{X_2}$ encapsulates the processed data, ready for further temporal modeling. To capture temporal dependencies, we introduce vector notations $\mathbf{x}_t$, $\mathbf{y}_t$, and $\mathbf{h}_t$, where $t$ indexes the position in the sequence. $\mathbf{x}_t \in \mathbb{R}^{d \times 1}$ represents the transpose of the $t$-th row vector of $\mathbf{X_2}$, while $\mathbf{h}_t \in \mathbb{R}^{d \times 1}$ and $\mathbf{y}_t \in \mathbb{R}^{1 \times d}$ denote the hidden state and output at time step $t$, respectively. The temporal dynamics are captured through a recursive update of the hidden state, which is formulated as:

$$\mathbf{h}_t = \mathbf{A}_t\mathbf{h}_{t-1} + \mathbf{b}_t\mathbf{x}_t, \tag{8}$$

where $\mathbf{A}_t \in \mathbb{R}^{d \times d}$ and $\mathbf{b}_t \in \mathbb{R}^{d \times 1}$ are matrices derived from linear transformations of the input $\mathbf{x}_t$. These matrices govern how the hidden state $\mathbf{h}_t$ evolves over time, incorporating both the previous hidden state $\mathbf{h}_{t-1}$ and the current input $\mathbf{x}_t$. The output at each time step $t$, denoted $\mathbf{y}_t$, is computed as:

$$\mathbf{y}_t = \mathbf{c}_t\mathbf{h}_t, \tag{9}$$

where $\mathbf{c}_t \in \mathbb{R}^{d \times 1}$ is another transformation matrix applied to the hidden state $\mathbf{h}_t$. By stacking the output vectors $\mathbf{y}_t$ across the time dimension, a matrix representing the final output of the encoder, denoted as $\mathbf{X}_{\text{enc}}$, is constructed. This matrix serves as the final encoded representation of the input sequence, encapsulating both local and temporal features for anomaly detection.

## 3.3 RECONSTRUCTOR

In the reconstruction phase, we receive $\mathbf{X}_{\text{enc}}$ from the encoder and then perform two operations: the first is patch partition, and the second is channel-independent linear reconstruction. These steps culminate in the generation of $\widehat{\mathbf{X}}$. The specific rationale behind this approach and the detailed transformation process can be found in Appendix D.

## 3.4 DETECTION

The reconstructed $\widehat{\mathbf{X}}$ is used differently in training and testing phases. During training, the objective is to minimize the reconstruction error, specifically the MSE between the input $\mathbf{X}$ and its reconstruction $\widehat{\mathbf{X}}$. The loss function $L$ is defined as:

$$L(\mathbf{X}, \widehat{\mathbf{X}}) = \frac{1}{n} \sum_{i=1}^{n} (\frac{1}{k} \sum_{j=1}^{k} (x_{ij} - \widehat{x}_{ij})^2), \tag{10}$$

where $n$ is the number of windows, $k$ is the number of features, and $x_{ij}$ and $\widehat{x}_{ij}$ are the true and reconstructed values, respectively. Minimizing this loss improves reconstruction fidelity and model performance. In testing, the anomaly score for the last point within a window is computed using the MSE for that point:

$$\text{Anomaly Score} = \frac{1}{k} \sum_{j=1}^{k} (x_{nj} - \widehat{x}_{nj})^2, \tag{11}$$

where $x_{nj}$ and $\widehat{x}_{nj}$ are the true and reconstructed values for the $n$-th (last) point in the window.

## 4 EXPERIMENTS

### 4.1 BENCHMARK DATASETS

We conducted comprehensive experimental comparisons across various datasets, including SMAP, MSL, SMD, PSM, and NIPS. For the NIPS dataset, we analyzed specific subsets: NIPS_TS_CCard, NIPS_TS_Swan, NIPS_TS_Syn_Mulvar, and NIPS_TS_GECCO, referred to as CCard, Swan, Mulvar, and GECCO, respectively. Statistical indicators of the dataset are provided in Appendix E.

### 4.2 BASELINES AND EVALUATION CRITERIA

We conducted a comprehensive evaluation of our model against 15 state-of-the-art baselines : TadGAN (Geiger et al., 2020) OmniAnomaly (Su et al., 2019), InterFusion (Li et al., 2021), THOC (Shen et al., 2020), Imdiffusion (Chen et al., 2023b), DiffAD (Xiao et al., 2023), ModernTCN (Luo & Wang, 2024), GDN (Deng & Hooi, 2021), TransAD (Tuli et al., 2022), MTAD-GAT (Zhao et al., 2020), Crossformer (Zhang & Yan, 2023), PatchTST (Nie et al., 2022), AnomalyTrans (Xu et al., 2021), DCdetector (Yang et al., 2023), itransformer(Liu et al., 2023) and TimeMixer++(Wang et al., 2024). We ensured fair comparison using metrics like precision, accuracy, and F1 on datasets such as SMD, MSL, SMAP, and PSM, and extended the evaluation with Affiliation metric (Huet et al., 2022) and VUS (Paparrizos et al., 2022) for newer datasets. Detailed metric descriptions are in Appendix F.

### 4.3 MAIN RESULTS

#### 4.3.1 PERFORMANCE

Experimental setup and environment are detailed in Appendix G. We first compare our model with classical and popular approaches across four standard benchmarks: SMD, MSL, SMAP, and PSM, focusing on precision, recall, and F1 score. The results are systematically presented in Table 1. Our model achieves the highest F1 scores on these datasets, consistently ranking among the top in precision and recall, demonstrating robustness in anomaly detection. We also compared it with AnomalyTransformer, DCdetector and DiffAD on NIPS datasets, as shown in Table 2. In Appendix L, we provide detailed sources of the results from the baseline model. KambaAD demonstrates

superior performance in F1 scores, particularly on the complex NIPS datasets, and exhibits clear advantages across multiple metrics. In the GECCO dataset, the KambaAD outperforms the other three models in accuracy and Aff-P but lags behind in Aff-R. This suggests KambaAD is conservative in anomaly detection, as it struggles to distinguish certain anomalies from normal patterns, resulting in similar anomaly scores for both. On the SWAN dataset, the other models show similar weaknesses, indicating that in datasets with a high diversity of anomaly patterns, all models exhibit weaknesses in detecting certain types of anomalies.

Table 1: Overall performance comparison of KambaAD and baseline models across four real-world multivariate datasets: SMD, MSL, SMAP, and PSM. Models are ranked from lowest to highest performance. Precision (P), Recall (R), and F1-score (F1) are reported in percentages (%). The best performance in each metric is highlighted in **bold**, and the second-best is underlined. A dash (-) indicates that the model's result is missing for the specific dataset.

| Dataset | SMD | | | MSL | | | SMAP | | | PSM | | |
|---|---|---|---|---|---|---|---|---|---|---|---|---|
| Metric | P | R | F1 | P | R | F1 | P | R | F1 | P | R | F1 |
| TadGAN | – | – | – | 89.02 | 86.37 | 62.30 | 92.49 | 81.99 | 70.40 | – | – | – |
| OmniAnomaly | 83.68 | 86.82 | 85.22 | 89.02 | 86.37 | 87.67 | 92.49 | 81.99 | 86.92 | 88.39 | 74.46 | 80.83 |
| InterFusion | 87.02 | 85.43 | 86.22 | 81.28 | 92.70 | 86.62 | 89.77 | 88.52 | 89.14 | 83.61 | 83.45 | 83.52 |
| THOC | 79.76 | 90.95 | 84.99 | 88.45 | 90.97 | 89.69 | 92.06 | 89.34 | 90.68 | 88.14 | 90.99 | 89.54 |
| ImDiffusion | 95.20 | 95.09 | 94.88 | 89..30 | 96.38 | 87.79 | 87.71 | 96.18 | 91.75 | 98.11 | 97.53 | 97.81 |
| DiffAD | 90.01 | 95.67 | 92.75 | 92.97 | 95.44 | 94.19 | 96.52 | 97.38 | 96.95 | 97.00 | **98.92** | 97.95 |
| ModernTCN | 87.86 | 83.85 | 85.81 | 83.94 | 85.93 | 84.92 | 93.17 | 57.69 | 71.26 | 98.09 | 96.38 | 97.23 |
| GDN | 71.70 | **99.74** | 83.42 | 93.08 | 98.92 | 95.91 | 74.80 | 98.91 | 85.18 | 87.50 | 83.85 | 85.64 |
| TranAD | 92.62 | **99.74** | 96.05 | 90.38 | 99.99 | 94.94 | 80.43 | **99.99** | 89.15 | 95.06 | 89.51 | 92.20 |
| MTAD-GAT | 88.36 | 83.30 | 84.63 | 87.54 | 94.40 | 90.84 | 89.06 | 91.23 | 90.13 | 87.63 | 87.25 | 87.44 |
| Crossformer | 83.06 | 76.61 | 79.70 | 84.68 | 83.71 | 84.19 | 92.04 | 55.37 | 69.14 | 97.16 | 89.73 | 93.30 |
| PatchTST | 87.42 | 81.65 | 84.44 | 84.07 | 86.23 | 85.14 | 92.43 | 57.51 | 70.91 | 98.87 | 93.99 | 97.23 |
| iTransformer | 78.45 | 65.10 | 71.15 | 86.15 | 62.65 | 72.54 | 90.67 | 52.96 | 66.87 | 95.65 | 94.69 | 95.17 |
| TimesMixer++ | 88.59 | 84.50 | 86.50 | 89.73 | 82.23 | 85.82 | 93.47 | 60.02 | 73.10 | 98.33 | 96.90 | 97.60 |
| AnomalyTrans | 88.47 | 92.28 | 90.33 | 91.92 | 96.03 | 93.93 | 93.59 | 99.41 | 96.41 | 96.94 | 97.81 | 97.37 |
| DCdetector | 83.59 | 91.10 | 87.18 | 93.69 | 99.69 | 96.60 | 95.63 | 98.92 | 97.02 | 97.14 | 98.74 | 97.94 |
| KambaAD | **97.10** | 97.45 | **97.27** | **98.84** | **100.00** | **99.41** | **98.46** | 99.93 | **99.19** | **99.15** | 97.00 | **98.06** |

### 4.3.2 KAN FOR WINDOW INFORMATION CAPTURE

We compared window-based and single-point features in KAN across eight datasets (Table 4). Results consistently show window-based input outperforming, with higher precision, recall, and F1 scores. This approach enhances KAN's capacity to capture temporal dependencies and inter-feature relationships, improving multivariate time series predictions.

### 4.3.3 ABLATION EXPERIMENT

Our ablation study compares accuracy, precision, and F1 score across eight datasets. We first compare KambaAD with its individual components (Encoder and Reconstructor). Table 4 shows KambaAD consistently outperforms both, emphasizing the importance of their integration. The comparison results when increasing the number of encoder and reconstructor parameters to match the total number of parameters under the full KambaAD can be found in the appendix J. We then analyze specific components: KAN for initial anomaly detection, Attention for global features, and MAMBA for local patterns. From Table 5, we can conclude that in the Mulva dataset, KambaAD's performance is inferior to using only the Encoder, while in the GECCO dataset, KambaAD's performance is lower than using only the Reconstructor. This suggests that certain specific characteristics

Table 2: Multi-metric performance comparison of KambaAD, DCdetector, and AnomalyTransformer on the NIPS dataset. Aff-P and Aff-R denote the precision and recall for the affiliation metric. R_A_R (Range_AUC_ROC) and R_A_P (Range_AUC_PR) represent scores based on label transformation under the ROC and PR curves, respectively. V_ROC and V_RR correspond to the volumes under the ROC and PR curve surfaces. All results are reported in percentages (%). The best performance in each metric is highlighted in **bold**, and the second-best is underlined.

| Dataset | Method | Acc | P | R | F1 | Aff-P | Aff-R | R_A_R | R_A_P | V_ROC | V_PR |
|---|---|---|---|---|---|---|---|---|---|---|---|
| Ccard | Anomalylrans | **99.84** | 13.33 | 0.90 | 1.68 | 64.84 | 4.77 | 52.53 | 11.56 | 52.47 | 11.87 |
| | Dcdetector | 98.73 | 0.06 | 0.45 | 0.11 | 50.56 | **71.63** | 52.91 | 10.40 | 52.68 | 9.99 |
| | DiffAD | 99.59 | 1.07 | 1.79 | 1.34 | 49.14 | 38.23 | 52.50 | 11.26 | 52.43 | 10.72 |
| | KambaAD | 99.77 | **34.29** | **53.60** | **41.83** | **73.54** | 67.04 | **52.98** | **26.04** | **53.42** | **26.12** |
| SWAN | Anomalylrans | 84.57 | 90.71 | 47.43 | 62.29 | 58.45 | 9.49 | 86.42 | 93.26 | 84.81 | 92.00 |
| | Dcdetector | 85.94 | 95.48 | 59.55 | 73.35 | 50.48 | 5.63 | 88.06 | 94.71 | 86.25 | 93.50 |
| | DiffAD | 86.40 | **99.15** | 58.78 | 73.80 | 48.12 | 1.00 | 87.41 | 94.77 | 85.37 | 93.51 |
| | KambaAD | **89.79** | 86.75 | **81.12** | **83.84** | **84.41** | **57.17** | **89.67** | **94.82** | **88.66** | **93.99** |
| Mulvar | Anomalylrans | 79.60 | 66.29 | 14.45 | 23.73 | 54.07 | 10.43 | 99.98 | 99.99 | 95.97 | 96.62 |
| | Dcdetector | 75.92 | 41.61 | 23.88 | 30.34 | 52.55 | 21.40 | **100.00** | **100.00** | 95.96 | 95.99 |
| | DiffAD | 80.31 | 69.31 | 18.58 | 29.30 | 58.07 | 16.61 | 99.98 | 99.99 | 96.02 | 96.76 |
| | KambaAD | **87.33** | **73.60** | **65.90** | **69.54** | **78.25** | **48.47** | 99.98 | 99.99 | **96.86** | **97.50** |
| GECCO | Anomalylrans | 98.03 | 25.65 | 28.48 | 26.99 | 49.23 | 81.20 | 56.35 | 22.53 | 55.45 | 21.71 |
| | Dcdetector | 98.56 | 38.25 | **59.73** | 46.63 | 50.05 | **88.55** | **62.95** | 34.17 | **62.41** | 33.67 |
| | DiffAD | 98.83 | 45.12 | 49.45 | 47.19 | 64.17 | 62.35 | 54.88 | 29.51 | 55.75 | 30.48 |
| | KambaAD | **99.31** | **99.61** | 35.21 | **52.02** | **99.95** | 13.46 | 51.73 | **52.70** | 51.88 | **52.86** |

Table 3: Performance comparison between KAN (point) and the proposed KambaAD model (KAN window) across eight real-world multivariate datasets. Precision (P), Recall (R), and F1-score (F1) are reported in percentages (%). The best results are highlighted in **bold**.

| Dataset | KAN(point) | | | KAN(window) | | |
|---|---|---|---|---|---|---|
| | P | R | F1 | P | R | F1 |
| SMD | 94.52 | 95.84 | 95.84 | **97.10** | **97.45** | **97.27** |
| MSL | 95.60 | **100.00** | 97.75 | **98.84** | **100.00** | **99.41** |
| SMAP | 96.44 | **99.93** | 98.16 | **98.46** | **99.93** | **99.19** |
| PSM | 97.48 | 95.16 | 96.31 | **99.15** | **97.00** | **98.06** |
| CCARD | 29.53 | 47.75 | 36.49 | **34.29** | **53.60** | **41.83** |
| SWAN | **97.84** | 65.27 | 78.30 | 86.75 | **81.12** | **83.84** |
| MULVAR | **77.06** | 65.79 | **70.98** | 73.60 | **65.90** | 69.54 |
| GECCO | **99.61** | **35.21** | **52.02** | **99.61** | **35.21** | **52.02** |

or anomaly patterns in the datasets become difficult to detect when processed through both the Encoder and Reconstructor. This may be due to overfitting caused by the larger number of parameters in the complete network structure. Overall, the complete model exhibits superior comprehensive performance across all datasets compared to models with removed components, indicating that the complete model possesses stronger generalization capabilities and stability.

### 4.3.4 ORDER OF COMPONENTS

The positioning of components within KambaAD is crucial, as it determines the order in which various anomalies ar e detected. Therefore, we have conducted analysis and experiments on this aspect, and more detailed information can be found in Appendix H.

Table 4: Performance comparison between the Encoder-only, Reconstruction-only, and KambaAD across eight real-world multivariate datasets. Precision (P), Recall (R), and F1-score (F1) are reported in percentages (%). The best results are highlighted in **bold**, and the second-best results are underlined.

| Dataset | Encoder | | | Reconstructor | | | KambaAD | | |
|---|---|---|---|---|---|---|---|---|---|
| | **P** | **R** | **F1** | **P** | **R** | **F1** | **P** | **R** | **F1** |
| **SMAP** | 95.89 | 99.78 | 97.80 | 93.26 | 99.87 | 96.45 | **98.46** | **99.93** | **99.19** |
| **MSL** | 85.45 | 99.55 | 91.96 | 93.12 | **100.00** | 96.44 | **98.84** | 100.00 | **99.41** |
| **SMD** | 95.93 | 95.90 | 95.91 | 95.99 | 96.61 | 96.30 | **97.10** | **97.45** | **97.27** |
| **PSM** | 97.33 | 94.01 | 95.64 | 98.06 | 94.22 | 96.14 | **99.15** | **97.00** | **98.06** |
| **Ccard** | 30.14 | 48.20 | 37.09 | **58.46** | 17.12 | 26.48 | 34.29 | **53.60** | **41.83** |
| **SWAN** | 82.96 | 79.49 | 81.19 | **92.17** | 63.79 | 75.40 | 86.75 | **81.12** | **83.84** |
| **Mulvar** | 70.76 | **73.81** | **72.25** | **78.14** | 65.03 | 68.55 | 73.60 | 65.90 | 69.54 |
| **GECCO** | **99.61** | 35.21 | 52.02 | 51.52 | **67.53** | **58.45** | 99.61 | 35.21 | 52.02 |

Table 5: Performance comparison between KambaAD and five ablation study models across eight real-world multivariate datasets. Only the comparison results of the F1 score are presented. The best results are highlighted in **bold**, and the second-best results are underlined.

| Dataset | KAN | ATT | Mamba | KAN ATT | KAN MAMBA | KAN MAMBA MAMBA | ATT MAMBA | KambaAD |
|---|---|---|---|---|---|---|---|---|
| **SMAP** | 97.55 | 98.13 | 97.80 | 98.25 | 97.83 | 97.99 | 97.88 | **99.19** |
| **MSL** | 97.32 | 96.89 | 96.74 | 96.90 | 98.27 | 96.68 | 94.14 | **99.41** |
| **SMD** | 96.25 | 95.05 | 95.56 | 94.27 | 95.10 | 96.04 | 89.92 | **97.27** |
| **PSM** | 97.29 | 97.05 | 97.22 | 96.78 | 96.58 | 90.78 | 78.88 | **98.06** |
| **Ccard** | 38.52 | 37.06 | 38.22 | 36.33 | 38.72 | 35.48 | 39.05 | **41.83** |
| **SWAN** | 76.20 | 75.53 | 78.10 | 78.72 | 78.97 | 78.70 | 78.30 | **83.84** |
| **Mulvar** | 78.31 | 75.73 | **83.37** | 70.86 | 63.05 | 64.39 | 56.75 | 69.54 |
| **GECCO** | 51.97 | 50.05 | **52.34** | 51.71 | 52.02 | 52.02 | 52.02 | 52.02 |

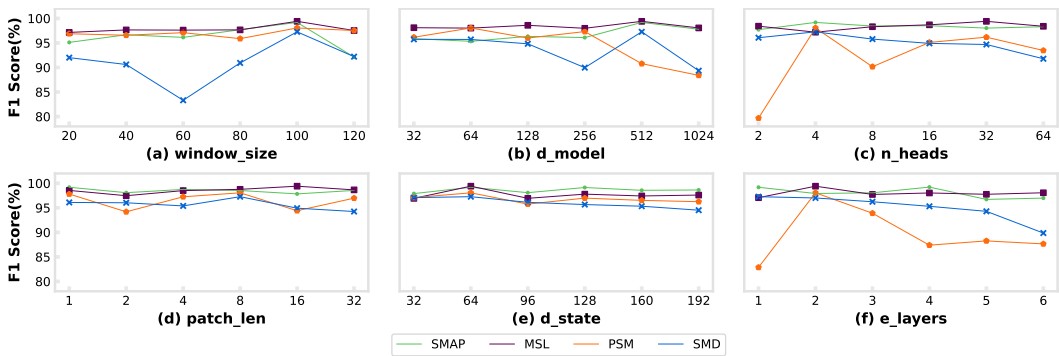

Figure 2: Parameter sensitivity studies of main hyper-parameters in KambaAD.

### 4.3.5 CHANNEL-INDEPENDENT (CI) OR CHANNEL-DEPENDENT (CD)

In this section, we compare the performance of CI and CD methods during reconstruction to support our previous analysis. As shown in Table 6, across all eight datasets, the CI approach consistently

outperforms CD, demonstrating superior reconstruction performance. The necessity of employing the CI reconstruction strategy lies in the observation that, during anomalies, some features are affected while others remain unaffected. To ensure that the reconstructed data is as normal as possible, different reconstruction strategies should be applied to these features. As shown in Figure 3, the behavior of feature 23 provides a clear example. In the encoder, we use the CD strategy, which causes feature 23 to be influenced by drastically changing features, such as features 12 and 16, after step 2. This contradicts the goal of reconstructing normal data. However, the final CI reconstruction successfully restores the data to a normal pattern.

Table 6: Performance comparison between channel-independent (CI) and channel-dependent (CD) reconstruction methods across eight real-world multivariate datasets. Precision (P), Recall (R), and F1-score (F1) are reported in percentages (%). The best results are highlighted in **bold**.

| Dataset | CD | | | CI(KambaAD) | | |
|---|---|---|---|---|---|---|
| | **P** | **R** | **F1** | **P** | **R** | **F1** |
| **SMAP** | 94.13 | **99.93** | 96.94 | **98.46** | **99.93** | **99.19** |
| **MSL** | 91.39 | **100.00** | 95.50 | **98.84** | **100.00** | **99.41** |
| **SMD** | 86.36 | 90.86 | 88.55 | **97.10** | 97.45 | **97.27** |
| **PSM** | 92.17 | 89.79 | 90.97 | **99.15** | **97.00** | **98.06** |
| **Ccard** | 29.97 | 46.85 | 36.56 | **34.29** | **53.60** | **41.83** |
| **SWAN** | **88.57** | 71.32 | 79.02 | 86.75 | **81.12** | **83.84** |
| **Mulvar** | 44.90 | 54.74 | 49.34 | **73.60** | **65.90** | **69.54** |
| **GECCO** | **99.61** | **35.21** | **52.02** | **99.61** | **35.21** | **52.02** |

### 4.3.6 PARAMETER SENSITIVITY

We conducted a sensitivity analysis on KambaAD, examining key parameters (window_size, patch_size, d_state, n_head, d_model, e_layers) and their impact on F1 scores across four datasets. As shown in Figure 2, KambaAD exhibits stability on the SMAP and MSL datasets, where parameter variations have a limited effect on performance. However, on the PSM and MSL datasets, the model is more sensitive to specific parameters. Notably, on the PSM dataset, setting n_heads and e_layers to 2 and 1 respectively leads to a significant performance drop, indicating that these parameter settings constrain the model's capabilities. For the SMD dataset, a window_size of 60 results in a noticeable decline in performance, suggesting that a larger context window is beneficial. Overall, KambaAD's performance remains stable, but further increasing hyperparameters such as d_model, n_heads, and e_layers does not enhance performance, likely due to overfitting.

### 4.3.7 VISUALIZATION

In the PSM dataset, we have approximately illustrated the data shapes after passing through the components KAN, Attention+MAMBA, and Reconstructor. It is important to note that this does not represent the data transformations within the complete KambaAD, as the data is projected into the model dimensions in KambaAD, making direct comparisons challenging. We incrementally built the model up to its full configuration and output the reconstructed data at each of these three structural stages, labeled as step 1, 2, and 3, to approximate the effects of each component. This analysis displays the anomaly scores and classifications (anomaly or normal) of each data point at each step. To enhance clarity, the PA strategy was omitted in this section. The chosen segment includes a point anomaly at the 10th step and a contextual anomaly around the 20th point. We visualized key features: feature 23, unrelated to anomalies; feature 4, related to anomalies near the 20th point; and features 12 and 16, related to both anomaly types. The results in Figure 3 show that the reconstructed data increasingly aligns with normalcy, detecting more anomalies. KAN's reconstruction shifts the original data to highlight obvious mutations, resulting in large errors for mutation points due to significant value differences, but it misses other anomalies. The Attention+MAMBA module incorporates contextual relationships, producing more coherent data and detecting more anomalies near the 20th point. However, it may cause excessive associations, such as unintended fluctuations in

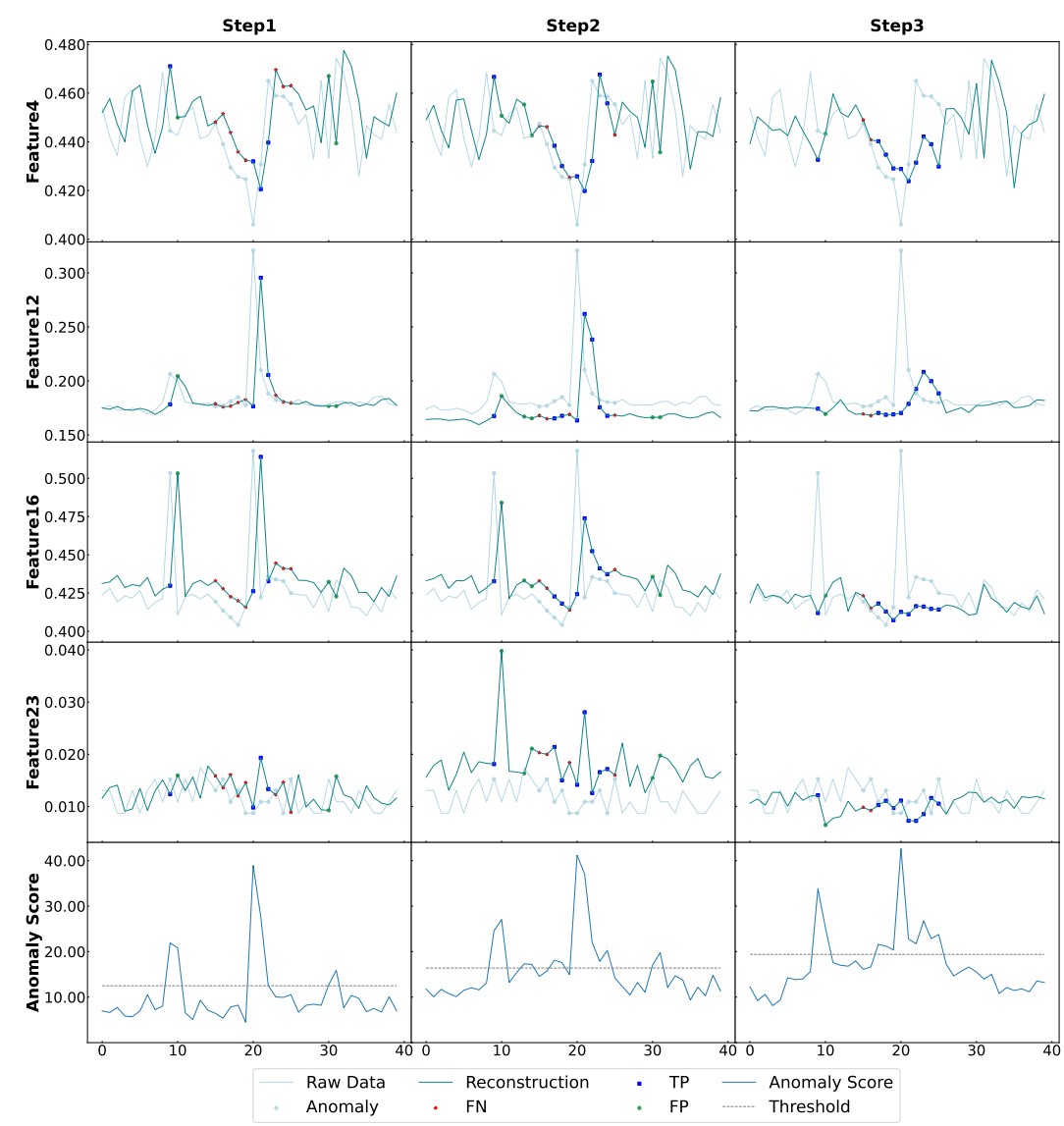

Figure 3: The presented figure illustrates the reconstruction of features 4, 12, 16, and 23 in a data sample from PSM following the extraction of three crucial components in KambaAD, along with the identification of anomalous points based on their reconstructed values using an optimal threshold.

feature 23 and deviations in normal data reconstruction. Finally, the Reconstructor normalizes features 4, 16, and 23, while feature 12 still reflects anomaly effects but sufficiently indicates anomalies through its divergence from the original data.

## 5 CONCLUSION

This paper introduced KambaAD, a novel encoder-reconstructor framework for time series anomaly detection. By integrating KAN for initial screening and a combined attention-MAMBA approach for refined detection, KambaAD effectively captures both global and local anomalies. Experimental results demonstrate that KambaAD achieves state-of-the-art performance, surpassing existing methods. Ablation studies further validate the contribution of each component to KambaAD's overall effectiveness. Future work will explore extending KambaAD to multivariate time series.

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

# A ALGORITHM

---

**Algorithm 1** KambaAD Model: Encoder and Reconstructor

---

**Require:** Raw windowed data $x \in \mathbb{R}^{B \times W \times F}$, where $B$ is the batch size, $W$ is the window size, and $F$ is the feature dimension.

**Ensure:** Reconstructed data $x_{\text{reconstructed}}$

  1: **Initialize:** Multiple Encoders and a Reconstructor

  2: **Encoder:**

  3: **for** each encoder layer **do**

  4:      $x_{\text{KAN}} \leftarrow (\texttt{KAN+Dropout})(x_{input})$          ▷ Preliminary anomaly detection

  5:      $x_{\text{proj}} \leftarrow \texttt{Linear}(x_{\text{KAN}})$          ▷ Projection of $F \rightarrow D$

  6:      $x_{\text{norm1}} \leftarrow \texttt{LayerNorm}(x_{\text{proj}})$          ▷ Normalization

  7:      $x_{\text{attn}} \leftarrow (\texttt{MHA+Dropout})(x_{\text{norm1}})$      ▷ Multi-head Attention for Global anomaly detection

  8:      $x_{\text{res1}} \leftarrow x_{\text{attn}} + x_{\text{norm1}}$          ▷ Residual connection

  9:      $x_{\text{norm2}} \leftarrow \texttt{LayerNorm}(x_{\text{res1}})$          ▷ Normalization

10:      $x_{\text{MAMBA}} \leftarrow (\texttt{Mamba+Dropout})(x_{\text{norm2}})$      ▷ Local anomaly detection using Mamba

11:      $x_{\text{res2}} \leftarrow x_{\text{MAMBA}} + x_{\text{norm2}}$          ▷ Residual connection

12:      $x_{\text{norm3}} \leftarrow \texttt{LayerNorm}(x_{\text{res2}})$          ▷ Normalization

13:      $x_{\text{enc}} \leftarrow \texttt{Linear}(x_{\text{norm3}})$          ▷ Projection of $D \rightarrow F$

14: **end for**

15: **Reconstructor:**

16: $x_{\text{perm}} \leftarrow \texttt{Permutation}^{(0,2,1)}(x_{\text{enc}})$          ▷ Preparation for patch unfolding

17: **if** $pad = \text{'end'}$ **then**

18:      $x_{\text{pad}} \leftarrow \texttt{ReplicationPad1d}(x_{\text{perm}})$          ▷ padding if necessary

19: **else**

20:      $x_{\text{pad}} \leftarrow x_{\text{perm}}$          ▷ No padding

21: **end if**

22: $x_{\text{patch}} \leftarrow \texttt{Unfold}^{(\texttt{dimension=-1},P,S)}(x_{\text{pad}})$          ▷ Patch division

23: $x_{\text{patch\_proj}} \leftarrow \texttt{Linear}(x_{\text{patch}})$          ▷ Increase dimension to $D$

24: $x_{\text{reconstructed}} \leftarrow \texttt{CI\_Linear}(x_{\text{patch\_proj}})$          ▷ Channel-independent reconstruction

25: $x_{\text{reconstructed}} \leftarrow \texttt{Permutation}^{(0,2,1)}(x_{\text{reconstructed}})$          ▷ Final output permutation

       **return** $x_{\text{reconstructed}}$

---

# B RELATED WORK: CLASSICAL, MODELS FOR ANOMALY DETECTION

Statistical methods, particularly effective for low-dimensional data, include moving averages, exponential smoothing, and the Autoregressive Integrated Moving Average (ARIMA) model (Box & Pierce, 1970). Moving averages smooth out short-term fluctuations in data to identify trends, while exponential smoothing gives more weight to recent observations. The ARIMA model combines autoregression, differencing, and moving averages to capture temporal dependencies in time series data. These models calculate residuals, where larger residuals may indicate anomalies (Yaacob et al., 2010). If the anomaly score, derived from these residuals, exceeds a specified threshold, the data is classified as anomalous.

Machine learning-based methods encompass a range of approaches, from classical algorithms to advanced deep learning techniques. Classical algorithms include:

- **One-Class Support Vector Machines (One-Class SVM):** This method identifies a boundary around normal data points, classifying points outside the boundary as anomalies.

- **k-Nearest Neighbor (k-NN):** It classifies data points based on their proximity to other points, with outliers being far from their neighbors.

- **Random Forests:** An ensemble method that constructs multiple decision trees, with outliers being those that frequently end up in the less common branches.

- **k-means clustering:** This method groups data into clusters, where points far from any cluster center are considered anomalies.

- **Gaussian Mixture Models:** These models assume that data is generated from a mixture of several Gaussian distributions, with anomalies being points that don't fit well into any distribution.
- **Isolation Forest:** It isolates anomalies by recursively partitioning the data, with anomalies being the first to be isolated.
- **Local Outlier Factor (LOF):** This method identifies anomalies by comparing the local density of each point to that of its neighbors.

Advanced deep learning techniques include:

- **Recurrent Neural Networks (RNN):** Designed for sequential data, RNNs capture temporal dependencies in time series (Hopfield, 1982).
- **Long Short-Term Memory (LSTM):** A special type of RNN, LSTMs are particularly effective at learning long-term dependencies in sequences (Hochreiter & Schmidhuber, 1997).
- **Autoencoders (AE):** These neural networks learn to encode data into a lower-dimensional space and then reconstruct it; anomalies are identified by high reconstruction errors (Rumelhart et al., 1986).
- **Variational Autoencoders (VAE):** A probabilistic variant of autoencoders that models data distributions and identifies anomalies through reconstruction errors or latent space deviations (Kingma & Welling, 2013).
- **Generative Adversarial Networks (GAN):** These consist of two networks (a generator and a discriminator) that learn to generate data; anomalies are identified by the discriminator's failure to classify generated data correctly (Goodfellow et al., 2014).
- **Transformers:** Known for their attention mechanisms, Transformers are effective at processing sequential data, especially in contexts where relationships between different parts of the sequence matter (Vaswani et al., 2017).
- **Graph Neural Networks (GNN):** These networks are tailored for data with graph structures, identifying anomalies based on the relationships between nodes in the graph (Kipf & Welling, 2016).
- **Diffusion models:** A newer approach that uses probabilistic methods to model complex data distributions and identify anomalies based on how well data fits these distributions.
- **Mamba:** Mamba is an innovative state space model that has recently gained attention in the field of machine learning and natural language processing. Developed as an alternative to traditional transformer architectures, Mamba leverages the power of state space models to process sequential data efficiently(Gu & Dao, 2023).

In anomaly detection, these methods generate a score for each data point, which is then compared to a threshold to determine whether the point is normal or anomalous.

Advanced deep learning methods are increasingly employed for anomaly detection due to their ability to capture complex patterns in high-dimensional data. Techniques such as Recurrent Neural Networks (RNN) (Choi et al., 2021), Long Short-Term Memory (LSTM) (Wei et al., 2023), Autoencoders (AE) (Thill et al., 2021), Variational Autoencoders (VAE) (Chen et al., 2023a), Generative Adversarial Networks (GAN) (Geiger et al., 2020), Transformers (Xu et al., 2021), Graph Neural Networks (GNN) (Deng & Hooi, 2021), Diffusion models (Wolleb et al., 2022) and Mamba (Chen et al., 2024; He et al., 2024) are prominent examples. These methods generate a score for each data point, which is then compared to a threshold to assess whether the point is anomalous or within the normal range.

## C  KAN ARCHITECTURE AND ITS IMPLEMENTATION

KAN, with its efficient parameterization and ability to approximate complex functions, effectively identifies anomalies that violate physical laws. The theoretical basis of KAN is the Kolmogorov-Arnold Representation Theorem, which states that any multivariate continuous function can be expressed as a composition of univariate functions and summation operations. Each univariate function

is modeled as a B-spline curve with learnable coefficients. In this section, we describe the custom implementation of the Kernel Activation Network (KAN). Unlike the original KAN, our model simplifies the input transformations, ensuring efficient information flow through the network. The steps are as follows:

## C.1 INPUT RESHAPING

Let the input tensor be $\mathbf{X} \in \mathbb{R}^{B \times T \times F}$, where:

- $B$ is the batch size,
- $T$ is the sequence length (time steps),
- $F$ is the feature dimension.

We first reshape $\mathbf{X}$ into a two-dimensional matrix:

$$\mathbf{X}' = \text{Reshape}(\mathbf{X}, [B \times T, F]). \tag{12}$$

This results in $\mathbf{X}' \in \mathbb{R}^{(B \times T) \times F}$, flattening the batch and time dimensions for further computation.

## C.2 KAN TRANSFORMATION

In a K-layer KAN (Kernel Activation Network), the transformation through the network is constructed as a series of operations applied across multiple layers. This can be represented mathematically as:

$$\text{KAN}(Z) = \left( \Phi_{K-1} \circ \Phi_{K-2} \circ \cdots \circ \Phi_1 \circ \Phi_0 \right) Z. \tag{13}$$

Here, $Z \in \mathbb{R}^{n_{\text{in}}}$ is the input vector to the network, and $\Phi_i$ represents the operation performed by the $i$-th layer. Each layer in the network receives an input of size $n_{\text{in}}$ and outputs a vector of size $n_{\text{out}}$.

Every KAN layer $\Phi_i$ comprises a set of learnable activation functions denoted by $\phi_{q,p}$, where each function $\phi_{q,p}$ maps input dimension $p$ to output dimension $q$, with $p = 1, 2, \ldots, n_{\text{in}}$ and $q = 1, 2, \ldots, n_{\text{out}}$. The entire transformation performed by each layer $k$ can thus be described as:

$$Z_{k+1} = \Phi_k Z_k. \tag{14}$$

where $Z_k \in \mathbb{R}^{n_k}$ is the input to the $k$-th layer, and $Z_{k+1}$ represents the output. The transformation matrix $\Phi_k$ includes the learnable activation functions in the following form:

$$\Phi_k = \begin{pmatrix} \phi_{k,1,1}(\cdot) & \phi_{k,1,2}(\cdot) & \cdots & \phi_{k,1,n_k}(\cdot) \\ \phi_{k,2,1}(\cdot) & \phi_{k,2,2}(\cdot) & \cdots & \phi_{k,2,n_k}(\cdot) \\ \vdots & \vdots & \ddots & \vdots \\ \phi_{k,n_{k+1},1}(\cdot) & \phi_{k,n_{k+1},2}(\cdot) & \cdots & \phi_{k,n_{k+1},n_k}(\cdot) \end{pmatrix}. \tag{15}$$

Each element $\phi_{k,q,p}$ is a learnable activation function that governs the relationship between input feature $p$ and output feature $q$.

## C.3 NESTED TRANSFORMATION IN KAN

The entire KAN network can be viewed as a recursive application of these layer transformations, where each subsequent layer takes the output of the previous layer as input. Mathematically, this is expressed as:

$$Z_K = \Phi_{K-1} \circ \Phi_{K-2} \circ \cdots \circ \Phi_0 Z. \tag{16}$$

At every layer, the transformation matrix $\Phi_k$ modifies the input vector $Z_k$ to generate the output vector $Z_{k+1}$, where each activation function is applied to the corresponding elements of the input.

## C.4 RESTORING THE ORIGINAL SHAPE

The transformed output $\mathbf{K}$ is reshaped back to its original three-dimensional form:

$$\mathbf{K}' = \text{Reshape}(\mathbf{K}, [B, T, D]). \tag{17}$$

This operation restores the batch and sequence dimensions, which results in $\mathbf{K}' \in \mathbb{R}^{B \times T \times D}$, and $D$ is the dimension of the new feature space.

## C.5 RESIDUAL ADDITION AND DROPOUT

At this stage, we form a residual connection by combining the transformed output $\mathbf{K}'$ with a linear transformation of the original input. This process is formalized as:

$$\mathbf{Y} = \text{Dropout}(\mathbf{K}') + \mathbf{WX}. \tag{18}$$

In the above formula, $\text{Dropout}(\mathbf{K}')$ applies dropout regularization to the transformed output, enhancing the model's robustness and preventing overfitting. The term $\mathbf{WX}$ represents a linear transformation of the original input $\mathbf{X} \in \mathbb{R}^{B \times T \times F}$, with $\mathbf{W} \in \mathbb{R}^{F \times D}$ being a learnable weight matrix that projects the input into the new feature space of dimension $D$. This residual connection facilitates the flow of information from earlier layers, mitigating the vanishing gradient problem and enabling the network to learn both transformed and original features effectively.

## C.6 LAYER NORMALIZATION

Finally, the result $\mathbf{Y}$ is normalized using layer normalization:

$$\mathbf{Z} = \text{LayerNorm}(\mathbf{Y}). \tag{19}$$

This ensures that each feature in $\mathbf{Y}$ has a consistent scale, stabilizing training and improving convergence.

# D LINEAR RECONSTRUCTION

The primary objective of implementing patch segmentation lies in harnessing the inherent local similarities within time series data to efficiently extract coherent features from adjacent data points. By partitioning the data into smaller, manageable patches, we not only simplify the analysis process but also significantly reduce the computational complexity of subsequent reconstruction tasks. This approach essentially treats each patch as a self-contained entity, fostering a more streamlined processing pipeline where each patch's data is considered and manipulated as a unified whole. As a result, the overall reconstruction effort becomes more efficient and manageable, as the complexity of operations is distributed across smaller, more manageable segments. The reason we choose to perform patch partition after feature extraction is that anomalies often appear continuously. If the number of anomalous points within a patch exceeds that of normal points, it becomes difficult to reconstruct them into normal data. However, our goal is to make the reconstructed data as normal as possible, so that we can better distinguish points through errors.

Given an input data matrix $\mathbf{X}_{\text{enc}}$ processed by the encoder, where $\mathbf{X}_{\text{enc}}$ has dimensions $n \times k$, with $n$ representing the length of the data sequence and $k$ representing the feature dimension of each data point. Subsequently, this data matrix undergoes a process called *patching*, resulting in a new matrix $\mathbf{X}_{\text{patch}}$ with dimensions $\lceil \frac{n}{\text{patch\_len}} \rceil \times \text{patch\_len} \times k$, where patch_len is the predefined number of elements in each patch and $\lceil \cdot \rceil$ denotes the ceiling function.

During this transformation, the data sequence is uniformly divided into segments of length patch_len. If the length $n$ of the original data sequence $X_{\text{enc}}$ is not divisible by patch_len, the last patch will have a length less than patch_len. In such cases, a zero-padding strategy is employed to ensure that all patches have a uniform length of patch_len, maintaining the regularity of the matrix $\mathbf{X}_{\text{patch}}$ for subsequent processing or analysis.

At last, we conduct linear reconstruction in an independent channel manner. In multivariate time series, each feature exhibits distinct trends, periodicity, seasonality, and other characteristics, and when anomalies occur, it does not necessarily mean that all features exhibit anomalies. Therefore,

to avoid mutual constraints and interference among features, we choose an independent channel approach. Given the data matrix $\mathbf{X}_{\text{patch}}$ with dimensions $\left\lceil \frac{n}{\text{patch\_len}} \right\rceil \times \text{patch\_len} \times k$, where $k$ represents the number of features, and $\left\lceil \frac{n}{\text{patch\_len}} \right\rceil$ is defined as patch_num, the number of patches. Our objective is to undertake a channel-independent linear reconstruction process, detailed as follows:

First, we execute a Channel-wise linear projection where, for each feature dimension $i \in \{1, 2, \ldots, k\}$, a unique linear transformation $\mathbf{W}_i \in \mathbb{R}^{\text{d\_model} \times \text{patch\_len}}$ is applied to project the patches $\mathbf{X}_{\text{patch}(\cdot,\cdot,i)}$ of the $i$-th feature from patch_len dimensions to d_model dimensions. This projection is mathematically expressed as:

$$\mathbf{Y}_i = \mathbf{X}_{\text{patch}(\cdot,\cdot,i)} \cdot \mathbf{W}_i^\top, \tag{20}$$

where $\mathbf{Y}_i \in \mathbb{R}^{\text{patch\_num} \times \text{d\_model}}$ represents the projected matrix with dimensions patch_num $\times$ d_model.

Subsequently, we perform a reshape operation by transforming each $\mathbf{Y}_i$ into a column vector $\mathbf{y}_i \in \mathbb{R}^{(\text{patch\_num} \cdot \text{d\_model}) \times 1}$ to facilitate further processing.

Next, we undertake a final dimensionality reduction step where, for each $\mathbf{y}_i$, a linear transformation $\mathbf{V}_i \in \mathbb{R}^{n \times (\text{patch\_num} \cdot \text{d\_model})}$ is applied to reduce the dimensionality from (patch_num $\cdot$ d_model) to $n$, yielding $\widehat{\mathbf{x}}_i \in \mathbb{R}^{n \times 1}$:

$$\widehat{\mathbf{x}}_i = \mathbf{V}_i \mathbf{y}_i. \tag{21}$$

Finally, we execute a concatenation process to form the reconstructed matrix $\widehat{\mathbf{X}} \in \mathbb{R}^{n \times k}$ by concatenating all $\widehat{\mathbf{x}}_i$ (for $i = 1, 2, \ldots, k$) along the feature dimension. Each column of $\widehat{\mathbf{X}}$ represents the reconstructed values for a corresponding feature.

This process completes the transformation from the original patch representation to the reconstructed time-series data, while preserving the independence of features.

# E   DATASET STATISTICS

The statistics of all datasets are illustrated in the Table 7.

Table 7: Dataset Statistics.

| Benchmark | Dimension | #Training | #Test (Labeled) | AR (%) |
|---|---|---|---|---|
| MSL(Mars Science Laboratory dataset) | 55 | 58, 317 | 73, 729 | 10.5 |
| NIPS_TS_Ccard | 28 | 142, 403 | 142, 404 | 0.2 |
| NIPS_TS_Swan | 38 | 60, 000 | 60, 000 | 32.6 |
| NIPS_TS_Syn_Mulvar | 5 | 80, 000 | 80, 000 | 22 |
| NIPS_TS_GECCO | 9 | 69, 260 | 69, 261 | 1.1 |
| PSM(Pooled Server Metrics) | 25 | 132, 481 | 87, 841 | 27.8 |
| SMAP(Soil Moisture Active Passive dataset) | 25 | 138, 004 | 435, 826 | 12.8 |
| SMD(Server Machine Dataset) | 38 | 708, 405 | 708, 420 | 4.2 |

# F   METRICS

## F.1   AFFILIATION METRIC: A METRIC FOR COMPREHENSIVE EVENT LOCALIZATION ASSESSMENT

The Affiliation metric, introduced by Huet et al., represents a sophisticated metric that integrates both precision and recall to evaluate the accuracy of event localization in a manner that is robust to potential interferences. This metric innovatively utilizes the Hausdorff distance to measure the

disparity between the true and predicted events, thereby providing a comprehensive assessment of the spatial alignment between the two. Furthermore, it incorporates individual probabilities within the designated Affiliation region for normalization purposes, enhancing the metric's sensitivity to varying degrees of confidence in the predictions.

A notable strength of the Affiliation metric lies in its creative application of the Hausdorff distance, which is known for its effectiveness in quantifying the maximum mismatch between two sets of points. This characteristic enables the metric to capture fine-grained discrepancies in event localization, offering a nuanced perspective on the performance of the system. Additionally, the integration of probabilities within the Affiliation region ensures that the score reflects not only the spatial accuracy but also the level of certainty associated with each prediction.

However, it is crucial to acknowledge the limitations of the Affiliation metric as well. Firstly, the size of the Affiliation region exerts a significant influence on the resulting score, potentially leading to an overestimation of performance when minimal gains in precision are achieved. This highlights the need for careful calibration of the region's dimensions to ensure an unbiased evaluation. Secondly, the metric exhibits a limitation in discriminating between false predictions within the Affiliation region, potentially masking errors that would otherwise be revealed. Lastly, the Affiliation metric exhibits a bias towards false positives over false negatives, which may skew the overall assessment of the system's performance, particularly in scenarios where a high degree of accuracy is paramount.

To mitigate these limitations, future research could explore the refinement of the Affiliation region's definition, as well as the development of additional metrics that complement the Affiliation metric in capturing different aspects of event localization performance. By addressing these challenges, the Affiliation metric can be further strengthened as a valuable tool for assessing and comparing the accuracy of event localization systems in scientific research.

## F.2 THE VOLUME UNDER THE SURFACE (VUS) METRIC: ENHANCING ANOMALY DETECTION EVALUATION THROUGH DISTANCE-BASED INSIGHTS

The Volume Under the Surface (VUS) metric, introduced by Paparrizos et al., represents a groundbreaking extension of AUC-based evaluation methodologies, specifically tailored to accommodate distance-based anomalies. Its fundamental novelty stems from the innovative label transformation technique employed, coupled with the meticulous computation of the volumetric aspect beneath the ROC curves constructed across a spectrum of buffer lengths. This intricate approach transcends traditional binary labeling, transforming it into a continuum of values that inherently biases towards an overestimation of false positives compared to false negatives, thereby providing a more nuanced and informative view of anomaly detection performance.

By seamlessly integrating this sophisticated label transformation mechanism with a meticulous volumetric assessment beneath the ROC surface, the VUS metric presents a comprehensive and multifaceted evaluation framework for anomaly detection systems. This framework is particularly adept at capturing nuances in performance that are often overlooked by conventional metrics, particularly in scenarios where the proximity to the decision boundary is of paramount importance. By enabling a deeper understanding of how anomaly detection algorithms behave across varying levels of confidence and proximity to the threshold, the VUS metric empowers researchers and practitioners alike to assess and benchmark the performance of diverse anomaly detection techniques with unprecedented precision and rigor.

Furthermore, the VUS metric underscores the importance of considering not just the absolute classification accuracy but also the confidence associated with each prediction, as well as the distribution of predictions across the ROC space. This holistic approach enables a more complete and accurate portrayal of anomaly detection performance, ultimately facilitating the development and refinement of more effective and reliable anomaly detection systems. In summary, the VUS metric stands as a valuable and indispensable tool in the ongoing pursuit of enhancing anomaly detection capabilities within the scientific community.

## G   EXPERIMENTAL SETUP AND ENVIRONMENT

Our experiments were conducted on 4 A800 GPUs. In the course of our experiments, the model parameters exhibited variability in their configuration across diverse datasets. The same parameters will be applied to different data sets for presentation in Table 8 , while different parameters will be set for various data sets displayed in Table 9. Overall, for datasets with higher feature dimensions, we require larger network architectures. For instance, MSL, SMD, and SWAN have the highest feature dimensions among all datasets, with 55, 38, and 38 dimensions respectively. Therefore, their hyperparameter settings are larger, with the d_model set to 512 in these three datasets. Conversely, for datasets with lower dimensions such as GECCO and Mulvar, which have 9 and 5 features respectively, the d_model only needs to be set to 64 and 128. Additionally, the stride and e_layers can almost always be set to 4 and 2, respectively, to achieve excellent results.

Table 8: The common hyperparameter settings used for training the model across all datasets.

| hyper-parameter | Value | hyper-parameter | Value |
|---|---|---|---|
| window_size | 100 | expand | 2 |
| batch_size | 8 | fc_dropout | 0.05 |
| dropout | 0.3 | d_conv | 4 |
| padding_patch | end | epochs | 2 |
| individual | 1 | d_state | 64 |

Table 9: The dataset-specific hyperparameter settings used for training the model on different datasets.

| Dataset | hyper-parameter | | | | |
|---|---|---|---|---|---|
| | patch_len | n_heads | d_model | stride | e_layers |
| **MSL** | 16 | 32 | 512 | 4 | 2 |
| **SMAP** | 1 | 2 | 512 | 8 | 1 |
| **SMD** | 8 | 4 | 512 | 4 | 2 |
| **PSM** | 8 | 4 | 64 | 4 | 2 |
| **Ccard** | 2 | 4 | 256 | 4 | 2 |
| **SWAN** | 2 | 8 | 512 | 4 | 2 |
| **Mulvar** | 1 | 32 | 128 | 4 | 2 |
| **GECCO** | 32 | 2 | 64 | 4 | 2 |

## H   ORDER OF COMPONENTS

In the KambaAD encoder, the sequence of the two components in the two-stage anomaly detection process is a crucial aspect of our design. This configuration is intentional: we aim for the KAN to capture evident *physical anomalies*, while the combination of attention and MAMBA is tasked with analyzing more subtle, less apparent anomalies. Specifically, the KAN is designed to detect anomalies that significantly deviate from expected physical properties, whereas attention+MAMBA component is engineered to identify more nuanced irregularities that might elude conventional detection methods. This hierarchical approach allows for a comprehensive anomaly detection process, addressing both obvious physical inconsistencies and intricate patterns that require more sophisticated analysis. Rigorous experiments compared KambaAD's encoder with configurations swapping these components. The results are shown in the Table 10 . Results show reversing the order reduced stability and accuracy, confirming KambaAD's design.

Our comprehensive experimental analysis provides strong evidence for the rationality and efficacy of the sequential order in KambaAD's two-stage anomaly detection process. The results consistently

Table 10: Performance comparison of three models with different component orders: KAN-MAMBA-attention, Attention-MAMBA-KAN, and KambaAD (KAN-attention-MAMBA) across eight real-world multivariate datasets. Precision (P), Recall (R), and F1-score (F1) are reported in percentages (%). The best results are highlighted in **bold**, and the second-best results are underlined.

| Dataset | KAN-MAMBA-attention | | | attention-MAMBA-KAN | | | KambaAD | | |
|---|---|---|---|---|---|---|---|---|---|
| | P | R | F1 | P | R | F1 | P | R | F1 |
| SMAP | 95.33 | 99.92 | 97.57 | 94.97 | 99.67 | 97.26 | **98.46** | **99.93** | **99.19** |
| MSL | 93.30 | 99.55 | 96.32 | 90.55 | **100.00** | 95.04 | **98.84** | **100.00** | **99.41** |
| SMD | 89.28 | 93.85 | 91.51 | 92.37 | 95.86 | 94.08 | **97.10** | **97.45** | **97.27** |
| PSM | 69.21 | 94.20 | 79.79 | 98.59 | **98.10** | **98.34** | **99.15** | 97.00 | 98.06 |
| Ccard | 31.27 | 45.50 | 37.06 | **59.68** | 16.67 | 26.06 | 34.29 | **53.60** | **41.83** |
| SWAN | **96.52** | 65.11 | 77.76 | 89.25 | 72.21 | 79.83 | 86.75 | **81.12** | **83.84** |
| Mulvar | 53.92 | 58.17 | 55.97 | **80.59** | **74.54** | **77.44** | 73.60 | 65.90 | 69.54 |
| GECCO | **99.61** | **35.21** | **52.02** | 99.23 | 35.21 | 51.97 | **99.61** | **35.21** | **52.02** |

show that this carefully designed sequence significantly enhances the overall performance of the model across various datasets and anomaly types.

# I  COMPARISON OF VISUAL RESULTS

This section visually compares the anomaly scores produced by KambaAD, DCdetector, and AnomalyTransformer on a segment of the NIPS_TS_Syn_Mulvar dataset with five distinct features. As shown in Figure 4, the comparison highlights each model's ability to detect anomalies in this complex time series. KambaAD demonstrates greater sensitivity, identifying subtle anomalies that DCdetector and AnomalyTransformer miss, particularly in regions where deviations are less apparent. This underscores KambaAD's effectiveness in capturing a wider range of anomaly patterns.

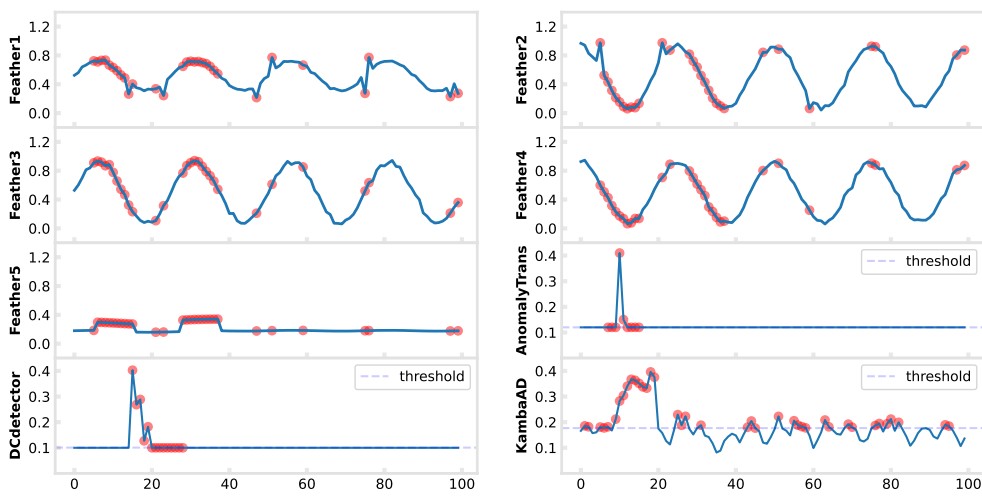

Figure 4: Comparison of anomaly scores from KambaAD, DCdetector, and AnomalyTransformer on the same data segment. The upper panel shows time series features with anomalies in red, while the lower panel presents the models' anomaly scores, also highlighting detected anomalies in red.

## J  ABLATION STUDY

This section presents the comparative results of ablation experiments conducted by increasing the number of Encoder and Reconstructor parameters to match those under the full KambaAD. We control that the number of parameters is almost equal across the data sets, and the results are shown in the Table 11. The observation reveals that even when the number of model parameters is increased to match KambaAD, employing only the Encoder or Reconstructor still yields inferior results compared to KambaAD. The performance remains largely unchanged from before increasing the parameter count, which aligns with our findings in the parameter sensitivity study.

Table 11: Performance comparison between the Encoder-only, Reconstruction-only, and KambaAD models across eight real-world multivariate datasets, with the model sizes kept approximately equivalent. Precision (P), Recall (R), and F1-score (F1) are reported in percentages (%). The best results are highlighted in **bold**, and the second-best results are underlined.

| Dataset | Encoder | | | Reconstructor | | | KambaAd | | |
|---|---|---|---|---|---|---|---|---|---|
| | **P** | **R** | **F1** | **P** | **R** | **F1** | **P** | **R** | **F1** |
| **MSL** | 89.64 | **100.00** | 94.54 | 91.74 | **100.00** | 95.69 | **98.84** | 100.00 | **99.41** |
| **SMAP** | 96.64 | 99.70 | 98.14 | 94.59 | 99.70 | 97.08 | **98.46** | **99.93** | **99.19** |
| **SMD** | 84.01 | 88.80 | 86.33 | 87.68 | 94.62 | 91.02 | **97.10** | **97.45** | **97.27** |
| **PSM** | 98.73 | 96.98 | 97.85 | 97.93 | **98.22** | **98.07** | **99.15** | 97.00 | 98.06 |
| **Ccard** | 30.14 | 48.20 | 37.09 | **49.35** | 17.12 | 25.42 | 34.29 | **53.60** | **41.83** |
| **SWAN** | 89.29 | 70.97 | 79.08 | **91.86** | 63.79 | 75.30 | 86.75 | **81.12** | **83.84** |
| **Mulvar** | 62.30 | 66.24 | 64.07 | 67.75 | 61.36 | 64.39 | 73.60 | 65.90 | 69.54 |
| **GECCO** | 82.37 | **35.21** | 49.33 | 95.90 | **35.21** | 51.50 | **99.61** | 35.21 | **52.02** |

## K  COMPARISON OF COMPUTATIONAL RESOURCE EFFICIENCY

This section evaluates the computational efficiency of our proposed model, KambaAD, in comparison with two state-of-the-art baselines: AnomalyTransformer and DCdetector. We measure efficiency using several key metrics: Training Time (seconds), GPU Usage (MB), Memory Usage (MB), Model Size (MB), and Parameter Count (millions). The GPU and memory usage are recorded during training with a batch size of 256, while the training time reflects the duration required to process the entire dataset. Our evaluation is conducted across four benchmark datasets: MSL, SMAP, SMD, and PSM.

For this comparison, we configure KambaAD with the following hyperparameters: $e\_layer = 1$, $d\_model = 128$, and $n\_heads = 4$. Although these settings are not optimized for maximum performance, they produce satisfactory results. A detailed comparison of the efficiency metrics for the three models across the four datasets is provided in Table 12.

The experimental results indicate that KambaAD offers a relatively short training time, which is a significant advantage. While the parameter count and model size of KambaAD are larger than those of the other two models—up to ten times larger than AnomalyTransformer—they remain within an acceptable range. Furthermore, the GPU usage during training remains low, addressing concerns that hyperparameter settings could create a performance bottleneck.

## L  THE SOURCES OF THE RESULTS FROM THE BASELINE MODEL

The experimental results for various baseline models were sourced from multiple publications to ensure a comprehensive and fair comparison. The results of TadGAN are derived from the papers that proposed TadGAN and StackedPredictor (Baireddy et al., 2021). Results for ImDiffusion, DiffAD, DCdetector and TimeMixer++ were obtained directly from their respective original publications. For OmniAnomaly, InterFusion, THOC, and AnomalyTransformer, we extracted the results from the DCdetector paper. For itransformer, we extracted the results from TimeMixer++ paper. In

Table 12: Comprehensive Computational Efficiency Analysis of KambaAD, AnomalyTransformer, and DCdetector: A Comparative Study across MSL, SMAP, SMD, and PSM Datasets. Metrics include Training Time (seconds), GPU Expend (MB), Memory Expend (MB), Model Size (MB), and Parameter Count (millions).

| Dataset | Model_Name | Train_Time(s) | GPU_Expend(MB) | Mem_Expend(MB) | Parameters |
|---------|------------|---------------|----------------|----------------|------------|
| MSL | DCdetector | 499.81 | **3836** | 1480.47 | **890,935** |
| | Anomaly | **83.46** | 8626 | 3154.69 | 4,863,055 |
| | KambaAD | 110.29 | 5428 | **1243.56** | 37,986,211 |
| SMAP | DCdetector | 580.62 | 9140 | 1523.19 | **883,225** |
| | Anomaly | 134.38 | 8622 | 2928.94 | 4,801,585 |
| | KambaAD | **54.12** | **4590** | **1189.79** | 18,529,821 |
| SMD | DCdetector | 1316.99 | 8998 | 4369.54 | **867,366** |
| | Anomaly | 988.31 | 9052 | 4285.21 | 4,828,222 |
| | KambaAD | **781.13** | **4166** | **2756.77** | 26,961,790 |
| PSM | DCdetector | 126.09 | 9930 | 1269.93 | **894,745** |
| | Anomaly | 82.44 | 8176 | 2887.36 | 4,801,585 |
| | KambaAD | **50.52** | **4590** | **1256.94** | 18,529,821 |

the case of GDN and TranAD, we utilized a combination of sources. The results for SMD, MSL, and SMAP datasets were sourced from the TranAD paper, while the PSM dataset results were obtained from the ImDiffusion paper. Similarly, for MTAD-GAT, the MSL and SMAP results were taken from the original MTAD-GAT paper, whereas the SMD and PSM results were sourced from the ImDiffusion paper. For Crossformer, PatchTST, and ModernTCN, all results were extracted from the ModernTCN paper, providing a consistent basis for comparison among these models. Regarding AnomalyTransformer and DCdetector, we adopted a dual approach. The results for the NIPS_TS_Swan and NIPS_TS_GECCO datasets were sourced from the DCdetector paper. However, for the NIPS_TS_CCard and NIPS_TS_Syn_Mulvar datasets, we conducted our own experimental evaluations to ensure completeness and verify the models' performance under our specific experimental conditions.

