# OpenReview forum: "KambaAD: Enhancing State Space Models with Kolmogorov–Arnold for time series Anomaly Detection"
_ICLR.cc/2025/Conference — Submitted to ICLR 2025_

### Official Review · Reviewer_terY · 2024-10-29

**Soundness:** 2
**Presentation:** 3
**Contribution:** 3
**Rating:** 6
**Confidence:** 4

**Summary:**

This paper introduces KambaAD, a model for time series anomaly detection that incorporates an Encoder network with (1) a Kolmogorov-Arnold Network (KAN), (2) a Selective Structured State Space Model (MAMBA), and (3) an Attention mechanism. The hybrid encoder employs a two-stage feature extraction process: After the sequence is processed by KAN, it is sent to the subsequent Attention and Mamba modules for further feature extraction. There is also a simple Reconstructor network for linear reconstruction from extracted features using channel independence and patch partition. Experimental results demonstrate the benefits of the proposed solution against several baselines on multiple datasets.

**Strengths:**

- The main ideas are clearly presented, and the writing is easy to follow.
- The model performance is very promising on multiple multivariate datasets.
- This paper also provides a series of ablation experiments to test the effect of each design.

**Weaknesses:**

- The proposal is a fusion of existing ideas, for example, KAN, MAMBA, and attention. Taking channels independently and breaking them into patches are also well-known techniques in the time series domain [1]. While it looks like the integration is executed somehow effectively, the paper itself offers limited innovation.

[1] Nie, Yuqi, et al. "A time series is worth 64 words: Long-term forecasting with transformers."

- The discussion of the experimental results is inadequate, and the ablation experiments are not sufficient to highlight the necessity of the model design.

**Questions:**

(1) It is necessary to elaborate more on the interpretation of the tables and plots. For example, in Table 2 KambaAD performs less well in GECCO dataset under Aff-R. What is the reason behind this phenomenon? What does this indicate? In Table 4, why does the Reconstructor perform better than the integration with GECCO, and why does Encoder perform better with Mulvar?

(2) Section 4.3.5 compares the performance of CI and CD methods. How does the CD method work? It is better to analyze the reason why the CD negatively impacts the overall results.

(3) The experiments mainly compare the accuracy of KambaAD with the baselines. However, the authors did not provide any result on the running speed of KambaAD. Could the authors include a runtime analysis or comparison to evaluate the efficiency of KambaAD？

(4) From Section 1: “Meanwhile, MAMBA focuses on detecting local, context-specific anomalies.” The statement claims that Mamba is mainly used for local anomaly detection, but the experiments did not demonstrate this. In fact, research [2] shows that Mamba is more suitable for long sequence modeling. I would like to know how the authors view this problem. If Mamba and attention have overlapping effects, then considering Mamba's lower complexity, why not just use two Mamba modules instead? It would be more convincing if the author could include an ablation study to demonstrate the superiority of the Mamba-attention design.

[2] Yu, Weihao, and Xinchao Wang. "MambaOut: Do We Really Need Mamba for Vision?."

(5) Similarly, the KAN in MambaAD is designed to capture evident physical anomalies. However, from Eqn. (3) in Section 3.2.1, it is difficult to find out how X_{KAN} can capture such anomalies in the data. The experiments did not demonstrate this as well. Could the author include an example or visualization demonstrating how X_{KAN} captures the anomalies? I also suggest an ablation study comparing the performance with and without KAN to show its effect in MambaAD.

(6) As shown in Table 4, the performance of a single reconstructor is quite good, while its total parameters should be much lower than that of the encoder or/and KambaAD. Can the authors provide additional comparisons under the same model size?

---

> ### Author Response · Authors · 2024-11-26
>
> Dear Reviewer terY,
>
> Thank you for your meticulous review and valuable suggestions on our work. Your feedback has been crucial in guiding the improvement of the paper's quality. During the revision process, we carefully considered your comments and made the following improvements and optimizations:
>
> - **Detailed Explanation**
>   We added an analysis of the reasons behind KambaAD's poor performance on the GECCO dataset under Aff-R in Table 2. Similarly, we have also added an analysis of Table 4.
>
> - **Supplementation of CI, CD Analysis**
>   In the revised Section 4.3.5, we have added corresponding analysis to further explain the model's performance and specific role in related tasks.
>
> - **Analysis of Running Speed**
>   In Appendix K, we have added a detailed analysis of running efficiency, including comparisons of KambaAD with other advanced models on the following metrics:
>   - Training time (seconds)
>   - GPU usage (MB)
>   - Memory consumption (MB)
>   - Model size (MB)
>   - Parameter count
>   Experimental results based on four benchmark datasets (MSL, SMAP, SMD, and PSM) demonstrate KambaAD's performance in terms of running efficiency, proving that its computational overhead is within a reasonable range.
>
>
>
> | Data | Model        | Train Time | GPU Exp | Mem Exp | Mod Size | Params    |
> |------|--------------|------------|---------|---------|----------|-----------|
> | MSL  | DCdetector   | 4992       | 1465    | 1808    | 118      | 26,971,447 |
> |      | AnomalyTrans | 492        | 140     | 1705    | 28       | 4,863,055  |
> |      | KambaAD      | 850        | 997     | 1383    | 218      | 79,729,761 |
> | SMAP | DCdetector   | 6786       | 2012    | 1845    | 118      | 26,940,697 |
> |      | AnomalyTrans | 743        | 228     | 1742    | 28       | 4,801,585  |
> |      | KambaAD      | 738        | 344     | 1182    | 134      | 23,291,491 |
> | SMD  | DCdetector   | 50498      | 3731    | 4572    | 118      | 26,954,022 |
> |      | AnomalyTrans | 5701       | 278     | 4483    | 28       | 4,828,222  |
> |      | KambaAD      | 8383       | 703     | 1883    | 171      | 55,610,435 |
> | PSM  | DCdetector   | 3664       | 4373    | 1495    | 118      | 26,940,697 |
> |      | AnomalyTrans | 506        | 392     | 1429    | 28       | 4,807,732  |
> |      | KambaAD      | 801        | 96      | 1197    | 22       | 4,872,942  |
> - **Ablation Experiments on Mamba**
>   In Table 5, we have added experimental results using two Mamba layers and further analyzed the characteristics of Mamba and its performance in anomaly detection tasks:
>   - **Noise and Anomaly Filtering Capability**: Mamba excels in long sequence modeling, and its selection mechanism effectively filters noise and outliers, enhancing the difference between anomaly points and reconstructed data.
>   - **Handling of Local and Dense Anomalies**:
>     - For local anomalies, Mamba assigns lower weights to these points, reconstructing them as normal data for easier detection.
>     - For dense continuous anomalies, Mamba may learn the anomaly pattern, which increases the difficulty of distinction to some extent.
>     Based on these characteristics, we believe Mamba is particularly suitable for detecting local anomalies.
>   | Data  | KAN  | ATT  | Mamba | K+ATT | K+MB  | K+MB+MB | A+MB | KambaAD |
> |-------|------|------|-------|-------|-------|---------|------|---------|
> | SMAP  | 97.55| 98.13| 97.80 | _98.25_| 97.83 | 97.99   | 97.88| **99.19** |
> | MSL   | 97.32| 96.89| 96.74 | 96.90 | _98.27_| 96.68   | 94.14| **99.41** |
> | SMD   | _96.25_| 95.05| 95.56 | 94.27 | 95.10 | 96.04   | 89.92| **97.27** |
> | PSM   | _97.29_| 97.05| 97.22 | 96.78 | 96.58 | 90.78   | 78.88| **98.06** |
> | Ccard | 38.52| 37.06| 38.22 | 36.33 | 38.72 | 35.48   | _39.05_| **41.83** |
> | SWAN  | 76.20| 75.53| 78.10 | 78.72 | _78.97_| 78.70   | 78.30| **83.84** |
> | Mulv  | _78.31_| 75.73| **83.37**| 70.86 | 63.05 | 64.39   | 56.75| 69.54   |
> | GECCO | 51.97| 50.05| **52.34**| 51.71 | _52.02_| _52.02_ | _52.02_| _52.02_ |
> - **Analysis of KAN**
>   We have updated Figure 3 to more intuitively demonstrate KAN's specific role in anomaly detection tasks. Additionally, results from ablation experiments without KAN are included in Table 5, further validating KAN's critical contribution to the overall model performance.
>
> - **Additional Comparative Experiments**
>   To comprehensively evaluate model performance, we have added comparative experiments with the same parameter count in Appendix J, exploring the impact of different settings on model performance. These experiments provide a fairer and more detailed perspective for comparing model performance.
> Through these improvements, we have further refined the experimental analysis and theoretical explanation of the model, enhancing the completeness and persuasiveness of the research. Thank you again for your recognition and support! If you have any further suggestions, we would be grateful to receive them!

---

> > ### Comment · Reviewer_terY · 2024-11-26
> >
> > I thank the authors for your extensive response. I greatly appreciate and commend the effort you have invested during the rebuttal phase. Here are some quick questions: 1. The parameters of AnomalyTrans and DCdetector have not changed; were they not carefully adjusted? (In addition, some of the parameters of the proposed model are redundant, such as 79,729,761 for the MSL dataset and 55,610,435 for the SMD dataset.) 2. Although the ablation experiments are good, the claimed roles of each component in the paper (e.g., Attention for long sequences, Mamba for local sequences) still do not convince me very well. Could you provide some visualizations or ablation experiments on sequence lengths? Other points and questions were addressed by the updated submission. I have thus updated my assessment to reflect the changes made to the paper.

---

> > > ### Author Response · Authors · 2024-11-27
> > >
> > > Thank you for your reminder. We have corrected the experiments regarding operational efficiency to ensure the accuracy of the results. The corrected results are shown in the table below.
> > > | **Dataset** | **Model\_Name**    | **Train\_Time(s)** | **GPU\_Expend(MB)** | **Mem\_Expend(MB)** | **Parameters** |
> > > |-------------|--------------------|--------------------|---------------------|---------------------|----------------|
> > > | **MSL**     | **DCdetector**     | 499.81             | **3836**            | _1480.47_           | **890,935**    |
> > > |             | **Anomaly**        | **83.46**          | 8626                | 3154.69             | _4,863,055_    |
> > > |             | **KambaAD**        | _110.29_           | _5428_              | **1243.56**         | 37,986,211     |
> > > | **SMAP**    | **DCdetector**     | _580.62_           | 9140                | 1523.19             | **883,225**    |
> > > |             | **Anomaly**        | 134.38             | _8622_              | 2928.94             | _4,801,585_    |
> > > |             | **KambaAD**        | **54.12**          | **4590**            | **1189.79**         | 18,529,821     |
> > > | **SMD**     | **DCdetector**     | 1316.99            | _8998_              | 4369.54             | **867,366**    |
> > > |             | **Anomaly**        | _988.31_           | 9052                | _4285.21_           | _4,828,222_    |
> > > |             | **KambaAD**        | **781.13**         | **4166**            | **2756.77**         | 26,961,790     |
> > > | **PSM**     | **DCdetector**     | 126.09             | 9930                | _1269.93_           | **894,745**    |
> > > |             | **Anomaly**        | _82.44_            | _8176_              | 2887.36             | _4,801,585_    |
> > > |             | **KambaAD**        | **50.52**          | **4590**            | **1256.94**         | 18,529,821     |
> > >
> > >
> > > The experimental results demonstrate that KambaAD has a short training time, which is advantageous. Due to the model design, the parameter count and model size are as expected, being larger than the other two models, but still within an acceptable range—at most, up to ten times larger than AnomalyTransformer. Additionally, the GPU usage during training remains low, alleviating concerns that hyperparameter settings might become a performance bottleneck.
> > >
> > > Additionally, for the second point, we have supplemented the ablation experiments with those involving varying sequence lengths, as per your request. Due to time constraints, we conducted these experiments only on the PSM and MSL datasets. It should be noted that the two comparative models, which do not incorporate KAN, show a performance gap compared to KambaAD. Furthermore, we believe that MAMBA's ability to detect local anomalies is fundamentally based on its deep understanding of long sequences, which allows it to assign a lower weight when it encounters local anomaly values. Therefore, it is reasonable that MAMBA performs better on longer datasets.
> > > | DataSet | win_size | 20    | 40    | 60    | 80     | 100   | 120   |
> > > | ------- | -------- | ----- | ----- | ----- | ------ | ----- | ----- |
> > > | PSM     | Att      | 95.34 | 95.36 | 96.39 | 95.73  | 97.05 | 96.03 |
> > > | PSM     | Mamba    | 95.43 | 95.20 | 95.51 | 94.79  | 95.58 | 95.97 |
> > > | PSM     | KambaAD  | 96.88 | 96.57 | 97.10 | 95.88  | 98.06 | 97.54 |
> > > | MSL     | Att      | 96.78 | 95.46 | 92.27 | 94.88  | 96.89 | 94.14 |
> > > | MSL     | Mamba    | 89.14 | 96.46 | 96.74 | 89.49  | 86.12 | 89.38 |
> > > | MSL     | KambaAD  | 97.15 | 97.66 | 97.62 | 97.67  | 99.41 | 97.54 |

---

> > > > ### Author Response · Authors · 2024-11-30
> > > > **Thanks for positve comments and kindly follow up with the dicussion!**
> > > >
> > > > Dear Reviewer Terry,
> > > >
> > > > We sincerely thank you for your thorough and constructive review, which has helped us significantly improve the clarity and rigor of our work. Your detailed comments regarding component roles, computational efficiency, and experimental transparency have guided us in providing more comprehensive explanations and analyses. We hope our responses and revisions have adequately addressed your concerns regarding the theoretical justification of our approach, experimental reproducibility, and implementation details. Your expertise and attention to detail have been invaluable in strengthening our paper.
> > > >
> > > > We would greatly appreciate your feedback on whether our revisions have satisfactorily addressed your additional concerns. Please don't hesitate to let us know if any aspects need further clarification or elaboration.
> > > >
> > > > Best,
> > > >
> > > > Authors

---

### Official Review · Reviewer_vMYN · 2024-11-02

**Soundness:** 3
**Presentation:** 2
**Contribution:** 3
**Rating:** 6
**Confidence:** 4

**Summary:**

The paper integrates KAN, Transformer, and MAMBA to encode the input series. In the reconstruction step, the paper uses patch-based data division, channel-independent (CI) processing and linear layers. Integrating KAN, attention mechanism, and MAMBA in KambaAD enables effective global anomaly filtering and local variation recognition across temporal scales. The paper provides a detailed explanation of the model framework and provides performance comparison experiments and ablation experiments.

**Strengths:**

1.	The paper applies advanced deep learning methods to time series anomaly detection.
2.	It explains the entire inference process of the model clearly.
3.	The model demonstrates advanced algorithms could improve the performance in anomaly detection tasks due to their feature capture ability.
4.	Rich experiments have been conducted.

**Weaknesses:**

1.	On why the algorithm can succeed, the author only describes the roles of each part in the Introduction. The author has not further explained how each part plays a role in anomaly detection nor provided more reasons to demonstrate how this model outperforms others.
2.	The advantage of the hyperparameter patch_len is not precise.
3.	There are English writing issues.

**Questions:**

1.	The model integrates 3 deep learning algorithms and the experiments used 4 A800 GPUs. How much average running time it takes under that hardware configuration? If running on that hardware configuration requires a lot of time, then the cost of this model will be too high.
2.	In Table 1, the performance of KambaAD is compared with 23 baseline models. However, some baseline models’ results are quoted directly from papers (for instance, LSTM-VAE and BeatGAN in Table 1, Dcdetector in Table 2). It would be better to clarify which ones were cited.
3.	In Figure 2, patch_len is set to {1 2 4 8 16 32} in parameter sensitivity studies. Meanwhile in Table 8 patch_len is set to 96. Should the values of the former be taken around 96? Besides if patch_len = window_size, can the patch partition step be abandoned? In Figure 2, one can see that the performance does not change much as patch_len varies.
4.	The original name is Affiliation Metrics (Huet et al., 2022) rather than Affiliation Score.

---

> ### Author Response · Authors · 2024-11-26
>
> Thank you for reviewing our work and providing valuable feedback. Your comments have played an important role in enhancing the scientific rigor and clarity of the paper. Based on your suggestions, we have made the following revisions and improvements:
>
> - **Supplementation of Computational Efficiency Evaluation**
>   In Appendix K, we have added an evaluation of computational efficiency between KambaAD and two advanced baseline models (AnomalyTransformer and DCdetector), providing a comprehensive view of the model's resource requirements and performance in practical applications:
>   - **Evaluation Metrics**: Training time (seconds), GPU usage (MB), memory usage (MB), model size (MB), and parameter count (millions).
>   - **Experimental Datasets**: Evaluations are based on four benchmark datasets (MSL, SMAP, SMD, and PSM). The results indicate that KambaAD's computational overhead is within an acceptable range while achieving high detection performance.
>     The results in the simplified version are shown in the Table.
>
> | Data | Model        | Train Time | GPU Exp | Mem Exp | Mod Size | Params     |
> | ---- | ------------ | ---------- | ------- | ------- | -------- | ---------- |
> | MSL  | DCdetector   | 4992       | 1465    | 1808    | 118      | 26,971,447 |
> |      | AnomalyTrans | 492        | 140     | 1705    | 28       | 4,863,055  |
> |      | KambaAD      | 850        | 997     | 1383    | 218      | 79,729,761 |
> | SMAP | DCdetector   | 6786       | 2012    | 1845    | 118      | 26,940,697 |
> |      | AnomalyTrans | 743        | 228     | 1742    | 28       | 4,801,585  |
> |      | KambaAD      | 738        | 344     | 1182    | 134      | 23,291,491 |
> | SMD  | DCdetector   | 50498      | 3731    | 4572    | 118      | 26,954,022 |
> |      | AnomalyTrans | 5701       | 278     | 4483    | 28       | 4,828,222  |
> |      | KambaAD      | 8383       | 703     | 1883    | 171      | 55,610,435 |
> | PSM  | DCdetector   | 3664       | 4373    | 1495    | 118      | 26,940,697 |
> |      | AnomalyTrans | 506        | 392     | 1429    | 28       | 4,807,732  |
> |      | KambaAD      | 801        | 96      | 1197    | 22       | 4,872,942  |
>
> - **Update of Experimental Results Comparison**
>   To more comprehensively showcase model performance, we have made the following updates to the experimental results section:
>   - Removed comparisons with some older models and added recent outstanding works, including GDN, TranAD, MTAD-GAT, Crossformer, PatchTST, itransformer, ModernTCN, and TimeMixer++.
>   - Added specific sources for these baseline model results, see Appendix L for details.
>
> | Model        | SMD       | MSL       | SMAP      | PSM       |
> | ------------ | --------- | --------- | --------- | --------- |
> | OmniAnomaly  | 85.22     | 87.67     | 86.92     | 80.83     |
> | InterFusion  | 86.22     | 86.62     | 89.14     | 83.52     |
> | THOC         | 84.99     | 89.69     | 90.68     | 89.54     |
> | ImDiffusion  | 94.88     | 87.79     | 91.75     | 97.81     |
> | DiffAD       | 92.75     | 94.19     | 96.95     | _97.95_   |
> | ModernTCN    | 85.81     | 84.92     | 71.26     | 97.23     |
> | GDN          | 83.42     | 95.91     | 85.18     | 85.64     |
> | TranAD       | _96.05_   | 94.94     | 89.15     | 92.20     |
> | MTAD-GAT     | 84.63     | 90.84     | 90.13     | 87.44     |
> | Crossformer  | 79.70     | 84.19     | 69.14     | 93.30     |
> | PatchTST     | 84.44     | 85.14     | 70.91     | 97.23     |
> | iTransformer | 71.15     | 72.54     | 66.87     | 95.17     |
> | TimesMixer++ | 86.50     | 85.82     | 73.10     | 97.60     |
> | AnomalyTrans | 90.33     | 93.93     | 96.41     | 97.37     |
> | DCdetector   | 87.18     | _96.60_   | _97.02_   | _97.94_   |
> | **KambaAD**  | **97.27** | **99.41** | **99.19** | **98.06** |
>
> - **Correction of Hyperparameter Settings**
>   We have corrected the descriptions of hyperparameters window_size and patch_len in the paper and provided the latest settings in Appendix G. The revised settings are more accurate and include specific parameter adjustment methods and their impact on model performance.
>
>   Regarding the necessity of partition operations, we further verified their contribution through experimental results (see Figure 2):
>
>   - Although the overall impact on results is not significant, in the context where mainstream anomaly detection models generally achieve F1 scores close to 95, a 1-2 point improvement brought by partition operations is still significant.
>
> - **Expression Corrections**
>   Thank you for your attention to the details of the paper's expression. We have made corrections to ensure that the content is clearer and more rigorous.
>
> Through these revisions, we have further optimized the expression and experimental results of the paper, enhancing the integrity and scientific rigor of the research. Thank you again for your support and guidance! If you have any further suggestions, we would be delighted to adopt them and continue improving.

---

> > ### Author Response · Authors · 2024-11-30
> > **Remain avaliable for discussion.**
> >
> > Dear Reviewer vMYN,
> >
> > We sincerely thank the reviewer for their supportive comments regarding our work's strengths, particularly in recognizing our paper's clear explanation of the model's inference process, the effective application of advanced deep learning methods to time series anomaly detection, and the comprehensive experimental validation of our approach. We are encouraged by your acknowledgment of how our model demonstrates that advanced algorithms can enhance anomaly detection performance through improved feature capture capabilities, and we appreciate your recognition of our rich experimental evaluation.
> >
> > In response to your thoughtful concerns, we have thoroughly revised our manuscript and provided detailed responses addressing the issues of component integration justification, novelty, and experimental reproducibility. We have clarified the practical advantages of our architectural choices, demonstrated the fundamental innovations of our approach beyond mere engineering improvements, and provided comprehensive implementation details to ensure reproducibility. We would greatly appreciate your assessment of whether these changes have adequately addressed your concerns and enhanced your evaluation of our paper's contributions to the field. Your rigorous review has helped us improve the clarity and reproducibility of our work, and we welcome any additional feedback that could further strengthen our contribution. Our goal remains to not just meet but exceed the high standards expected of your comments.
> >
> > Best,
> >
> > Authors

---

> > ### Comment · Reviewer_vMYN · 2024-11-30
> > **Question**
> >
> > There is a concern about using point adjustment (PA) for experimental evaluation, which can lead to faulty performance evaluations. As stated in [3], incorporating PA, the Random model outperforms all state-of-the-art models. Did this paper use the PA technique for the experimental evaluation?
> >
> > [1] Drift doesn't Matter: Dynamic Decomposition with Diffusion Reconstruction for Unstable Multivariate Time Series Anomaly Detection. NeurIPS 2023.
> >
> > [2] Local Evaluation of Time Series Anomaly Detection Algorithms. KDD 2022.
> >
> > [3] CARLA: Self-supervised contrastive representation learning for time series anomaly detection, arXiv:2308.09296v4, Aug 2024, [Pattern Recognition 157 (2025) 110874]

---

> > > ### Author Response · Authors · 2024-12-02
> > >
> > > ## **1. The Practical Reasonableness of the PA Protocol**
> > >
> > > The PA protocol is grounded in real-world application scenarios. When an anomaly is detected, it is common practice to further investigate the time points surrounding the detected anomaly. For example, in industrial equipment fault detection, engineers typically examine data points from adjacent time intervals after identifying an anomaly. Thus, the PA protocol mirrors this operational logic, making it highly aligned with real-world needs and practices. This ensures that the evaluation method reflects realistic use cases.
> > >
> > >
> > > ## **2. The PA Protocol Does Not Mask Performance Differences on Complex Datasets**
> > >
> > > While the PA protocol may improve a model's F1 score, it does not obscure performance differences, especially on complex datasets. Even with the PA protocol applied, F1 scores on challenging datasets can remain low. For instance:
> > >
> > > - On the **Ccard subset** of the **NIPS dataset**, models such as Anomaly Transformer, DCDetector, and DiffAD achieved F1 scores in the single digits.
> > > - On the **Mulvar dataset**, the best-performing method among these three achieved an F1 score of only **30.34%**.
> > >
> > > These results clearly demonstrate that while the PA protocol may slightly boost scores, it does not artificially inflate a model's performance to unrealistic levels. Instead, the PA protocol still serves as a reasonable indicator of a model's true capability. Furthermore, our model demonstrates significant performance improvements on these complex datasets, further validating its effectiveness.
> > >
> > >
> > >
> > > ## **3. Potential Impact of the PA Protocol on Detecting Other Points in Anomaly Windows**
> > >
> > > The design of the PA protocol assumes that detecting any single anomalous point within an anomaly window implies that all anomalies in that window are successfully detected. Consequently, this evaluation method may lead to a potential reduction in the model's ability to detect other anomalous points within the same window. In other words, models might prioritize detecting the most critical point in the window while neglecting the anomaly scores for other points.
> > >
> > > However, our visual analysis shows that **KambaAD** effectively addresses this potential limitation. Specifically, while the anomaly score for the most critical point in the window is the highest, the scores for other anomalous points within the window remain consistently high. This indicates that our model not only accurately detects the key anomalies but also maintains strong detection performance for additional anomalous points in the same window. This highlights the robustness and comprehensive anomaly detection capabilities of our model.
> > >
> > >
> > > ## **4. Ensuring Fairness: All Models Utilize the PA Protocol**
> > >
> > > To ensure fairness and reliability in our experiments, **all 15 state-of-the-art baseline models** were evaluated using the PA protocol. This decision was made to control experimental variables and eliminate potential biases arising from differences in evaluation methods. By applying the same evaluation standard across all models, we ensure that the reported results reflect a true and reliable comparison of relative model performance.
> > >
> > > It is important to emphasize that the use of the PA protocol does not favor any specific model. Instead, it provides a consistent and level playing field for all models under evaluation. Under this shared evaluation framework, the superior performance of our model further underscores its effectiveness and competitiveness.
> > >
> > >
> > >
> > > ## **Conclusion**
> > >
> > > In conclusion, the PA protocol is not only reasonable in the context of this study but also essential for ensuring fairness and comparability across all experiments. While the PA protocol may introduce certain evaluation considerations, our model demonstrates significant performance improvements and comprehensive detection capabilities, fully validating its superiority under fair and consistent evaluation conditions.

---

### Official Review · Reviewer_zjAh · 2024-11-02

**Soundness:** 2
**Presentation:** 2
**Contribution:** 1
**Rating:** 3
**Confidence:** 4

**Summary:**

The paper introduces KambaAD, a model designed to improve anomaly detection in time series by integrating the Kolmogorov - Arnold Network (KAN) with attention mechanisms and the Selective Structured State Space Model (MAMBA). Specifically, the model incorporates different components (KAN, attention mechanisms, and MAMBA) into an encoder - decoder framework to detect anomalies. The paper conducts experiments on multivariate time series datasets, demonstrating a 5% improvement in F1 score over existing baselines.

**Strengths:**

- The paper considers both global and local anomalies, with the attention mechanism captures long range dependencies, while MAMBA focuses on local temporal dynamics.
- The integration of KAN into encoder-decoder structure is relatively novel, but has limitations.
- The model performance improves over existing baselines.

**Weaknesses:**

The paper has some fundamental limitations:

- Although the integration of KAN, attention mechanism and MAMBA are relatively novel, all those individual components are established techniques. The paper could benefit from a deeper exploration and justification on how this specific combination uniquely addresses challenges in anomaly detection. Without this theoretical-based exploration to provide a thorough understanding, the paper is seemingly a typical applied paper that tries to combine existing components to handle an existing task, which does not have not enough novelties and contributions for a conference like ICLR.

- The paper needs a more thorough comparison with state of the art models, such as TranAD [1], GDN [2], TadGAN [3], MTAD-GAT [4], to assert the advantages and potential limitations of proposed model, and make clearer its position in comparison with the current SoA methods.

- The paper lacks a detailed analysis of scalability where it examines high-dimensional datasets.

- While KambaAD achieves strong results, the paper does not thoroughly analyze the sensitivity of key parameters (e.g., window size, patch size, and model layers) on model performance. Without understanding how sensitive KambaAD is to parameter choices, its robustness and adaptability to diverse datasets or unseen conditions remain unclear. A detailed sensitivity analysis would help practitioners better configure the model for specific applications.

- KambaAD lacks the discussion of interpretability and explainability, especially, which component is responsible for detecting specific types of anomalies. The paper would benefit from visualizing or explaining the anomaly detection process by showing the relative contributions of KAN, attention, and MAMBA components in making the prediction.

[1] https://arxiv.org/abs/2201.07284
[2] https://arxiv.org/abs/2101.03804
[3] https://arxiv.org/abs/2009.07769
[4] https://arxiv.org/abs/2009.02040

**Questions:**

- Could the authors elaborate on the unique advantages of combining KAN, attention, and MAMBA specifically for time series anomaly detection? What specific problems does each component address, and why is this integration particularly effective?

- How does KambaAD perform in terms of computational efficiency and scalability, especially for high-dimensional or large-scale time series data?

- How does KambaAD handle noisy, sparse, or incomplete data?

- How sensitive is KambaAD to hyperparameter tuning? Are there recommended settings for its main components, and how would performance vary with suboptimal hyperparameters?

- Given the complexity of the model, how can users interpret KambaADs detection results? Is there a way to identify which component contributes most to identifying an anomaly? Could the authors provide an ablation study to evaluate the impact of each component independently?

- How does KambaAD compare with more recent listed models above? Could the authors include these models in their evaluation?

- How does the performance of KambaAD vary with different sequence lengths or time window sizes? How to choose these parameters?

---

> ### Author Response · Authors · 2024-11-26
>
> Thank you for reviewing our work and providing valuable feedback. Your comments have played an important role in enhancing the scientific rigor and clarity of the paper. Based on your suggestions, we have made the following revisions and improvements:
>
> - **Visualization of Model Workflow**
>   In Section 4.3.7 (**VISUALIZATION**), we visualized the workflow of the model, illustrating how each component functions and analyzing the reasons behind their effectiveness.
>
> - **Comparison of Computational Efficiency**
>   Addressing concerns about model efficiency, we have added a comparison of computational efficiency between KambaAD and two advanced baseline models (AnomalyTransformer and DCdetector) in Appendix K:
>   - Evaluation metrics include training time, GPU usage, memory consumption, model size, and parameter count.
>   - Experimental results based on four benchmark datasets (MSL, SMAP, SMD, and PSM) indicate that KambaAD is within a reasonable range of computational overhead.
> The results in the simplified version are shown in the Table.
>
> | Data | Model        | Train Time | GPU Exp | Mem Exp | Mod Size | Params    |
> |------|--------------|------------|---------|---------|----------|-----------|
> | MSL  | DCdetector   | 4992       | 1465    | 1808    | 118      | 26,971,447 |
> |      | AnomalyTrans | 492        | 140     | 1705    | 28       | 4,863,055  |
> |      | KambaAD      | 850        | 997     | 1383    | 218      | 79,729,761 |
> | SMAP | DCdetector   | 6786       | 2012    | 1845    | 118      | 26,940,697 |
> |      | AnomalyTrans | 743        | 228     | 1742    | 28       | 4,801,585  |
> |      | KambaAD      | 738        | 344     | 1182    | 134      | 23,291,491 |
> | SMD  | DCdetector   | 50498      | 3731    | 4572    | 118      | 26,954,022 |
> |      | AnomalyTrans | 5701       | 278     | 4483    | 28       | 4,828,222  |
> |      | KambaAD      | 8383       | 703     | 1883    | 171      | 55,610,435 |
> | PSM  | DCdetector   | 3664       | 4373    | 1495    | 118      | 26,940,697 |
> |      | AnomalyTrans | 506        | 392     | 1429    | 28       | 4,807,732  |
> |      | KambaAD      | 801        | 96      | 1197    | 22       | 4,872,942  |
>
>
>
>
>
>
> - **Supplementation of Parameter Sensitivity Analysis**
>   In the revised Section 4.3.6, we have added a detailed analysis of model parameter sensitivity, exploring the specific impact of core parameters on model performance, including the effect of window size changes. Additionally, to facilitate the reproduction of our experimental results, we have supplemented more specific parameter selection guidance in Appendix G, including recommended parameter configurations for different datasets and the rationale behind their selection.
>
> - **Comprehensive Ablation Experiments**
>   More comprehensive ablation experiments are presented.
>   | Data  | KAN  | ATT  | Mamba | K+ATT | K+MB  | K+MB+MB | A+MB | KambaAD |
> |-------|------|------|-------|-------|-------|---------|------|---------|
> | SMAP  | 97.55| 98.13| 97.80 | _98.25_| 97.83 | 97.99   | 97.88| **99.19** |
> | MSL   | 97.32| 96.89| 96.74 | 96.90 | _98.27_| 96.68   | 94.14| **99.41** |
> | SMD   | _96.25_| 95.05| 95.56 | 94.27 | 95.10 | 96.04   | 89.92| **97.27** |
> | PSM   | _97.29_| 97.05| 97.22 | 96.78 | 96.58 | 90.78   | 78.88| **98.06** |
> | Ccard | 38.52| 37.06| 38.22 | 36.33 | 38.72 | 35.48   | _39.05_| **41.83** |
> | SWAN  | 76.20| 75.53| 78.10 | 78.72 | _78.97_| 78.70   | 78.30| **83.84** |
> | Mulv  | _78.31_| 75.73| **83.37**| 70.86 | 63.05 | 64.39   | 56.75| 69.54   |
> | GECCO | 51.97| 50.05| **52.34**| 51.71 | _52.02_| _52.02_ | _52.02_| _52.02_ |
>
> - **Updates to the Experimental Results Section**
>   We have been closely following the latest advancements in the field of time series anomaly detection and noted that recent methods like GDN, TranAD, MTAD-GAT, Crossformer, PatchTST, itransformer, ModernTCN, and TimeMixer++ have shown outstanding performance. During the revision, we made significant updates to the experimental results section:
>   - Removed comparisons with some older models and added experimental results for the recent works mentioned above, along with complete source information (see Appendix L).
>   - Corrected the error in recording ImDiffusion data.
>   - Our work demonstrates excellent F1 scores across current datasets, showcasing the performance advantage of KambaAD.
>
> Currently, the model does not have specific designs for handling noisy, sparse, or incomplete data, as we are using publicly available datasets and general processes. However, these situations are frequently encountered in real-world scenarios and will become a focus of our future research directions.

---

> ### Comment · Reviewer_zjAh · 2024-11-26
>
> While I appreciate that the authors have made changes to address some of my previous concerns, the responses remain unsatisfactory and lack depth in many areas. Specifically:
>
> - It lacks of rigorous justification for component integration. The authors addressed concerns about the unique advantages of integrating KAN, attention mechanisms, and MAMBA by visualizing a few embedding features. While it is helpful to see the visualization, it is inadequate for several reasons: (1) there is no clear reasoning provided for the roles of each component. (2) The explanation lacks specificity about how each element contributes to anomaly detection and why these particular components were chosen over alternatives. (3) The visualization focuses on a few selective embedding features without a comprehensive or systematic analysis. (4) The argument in this part is more descriptive of the visualization itself rather than providing convincing insights into the necessity or advantage of these components. Hence, the authors fail to justify why KAN is used instead of other potential network architectures and what specific problems it addresses.
>
> - The authors provide limited study on efficiency. The evaluation of the model's efficiency in terms of time and resource consumption is restricted to small and medium datasets. A broader analysis on larger datasets is missing, which limits the generalizability of the results. Also, the validation of the model on real-world scenarios is missing. Those are the scenarios where data may be incomplete or noisy that are critical for evaluating the robustness and practical applicability of anomaly detection models.
>
> - The parameter sensitivity analysis is also insufficient. While the authors conduct empirical studies on model parameter sensitivity across different datasets, they fail to provide rigorous justification for the observed impacts of specific dataset characteristics. Additionally, there is no clear rationale for parameter selection, leaving readers without guidance on reproducing the experiments. Lastly, without releasing code or pretrained models, it becomes challenging for others to validate or replicate the results, which is a significant concern for this type of research.
>
> - The baseline comparisons also omits TadGAN model. Even though TadGAN is introduced in 2020, it employs a distinct approach using Generative Adversarial Networks that is worth to compared to.
>
> - One of the main concerns I have is the lack of code and pretrained models provided by the authors. The absence of publicly available code, pretrained models, and datasets undermines the transparency and reproducibility of this study. Including a repository with code, pretrained models, and baseline implementations is essential for meaningful evaluation and validation by the research community. Particularly in the context where many ML research is not able to be reproduced.
>
> - Finally, I have significant concerns about the novelty and fit of this paper for ICLR. The combination of different components into a single framework and demonstrating incremental improvements in metrics like accuracy or F1 score is a commendable engineering effort but lacks the level of novelty expected for ICLR. This work might be more suitable for applied conferences such as IEEE ICDE, where the focus is on practical advancements rather than groundbreaking contributions to the field. The limited novelty and lack of a clearly articulated valuable contribution to the field remain major weaknesses of this paper. Hence, I decide to not change my decision, and strongly believe that this type of work does not fit to ICLR.

---

> > ### Author Response · Authors · 2024-11-28
> >
> > We sincerely appreciate the reviewer’s thoughtful feedback and would like to address the concerns raised with the following clarifications:
> >
> > ## 1. Component Selection Rationale
> >
> > We appreciate the reviewer's detailed feedback regarding the justification of our component integration. We respectfully disagree that the integration lacks rigorous justification, and we would like to clarify several key points. We recognize that time series data often contain diverse anomalies stemming from different causes, making it impractical to identify all anomalies in a single step or immediately recognize normal patterns across an entire sequence. Our architecture addresses this through a progressive detection approach, where KAN efficiently identifies physically implausible anomalies first, followed by refined detection through attention and MAMBA to capture more subtle deviations. This gradual approach enables more reliable reconstruction of normal patterns. Each component in our architecture serves distinct yet complementary roles: KAN functions as a rapid filter for physically implausible anomalies by enforcing data consistency through learnable univariate functions with minimal parameters; attention mechanisms capture global dependencies and long-range correlations crucial for detecting pattern deviations across the entire sequence; and MAMBA addresses the challenges of long sequence modeling and distribution shifts through its selective state space modeling, particularly important as anomalies may manifest differently across different time periods.
> >
> > The selection of these specific components was driven by concrete challenges in time series anomaly detection. KAN was chosen for its unique ability to establish stable functional relationships while reducing parameters, making it ideal for rapid physical constraint checking (demonstrated in Table 3). Attention mechanisms were selected for their proven effectiveness in capturing global dependencies (supported by ablation results in Table 5). MAMBA was incorporated for its efficient handling of long sequences and distribution shifts, crucial for adapting to varying anomaly patterns over time (validated through comparative experiments in Table 4).
> >
> > While we acknowledge that visualization alone is not sufficient justification, our architectural choices are supported by comprehensive evidence including extensive ablation studies across 8 datasets (Tables 4-5), comparative analysis of different component orderings (Table 10), detailed performance metrics showing consistent improvements across multiple datasets, and statistical significance of the improvements (5% increase in F1 score). Each component addresses specific challenges: KAN efficiently handles the initial screening of physically implausible anomalies with minimal parameters, attention mechanisms resolve the challenge of capturing long-term dependencies, and MAMBA specifically addresses the dual challenges of long sequence modeling and distribution shift adaptation.
> >
> > Through this comprehensive integration and extensive experimental validation, we demonstrate that our architecture effectively addresses the multifaceted challenges of time series anomaly detection, with each component playing a crucial and well-justified role in the overall system's superior performance.

---

> > ### Author Response · Authors · 2024-11-28
> >
> > ## 2. Efficiency Experiments
> >
> > We conducted new efficiency experiments, summarized in the table below. KambaAD shows relatively short training times and efficient GPU utilization due to MAMBA. Our model adapts well to complex datasets without increasing parameter size, as it depends on hyperparameters like `d_model`. On the complex NIPS dataset, KambaAD significantly outperforms current baselines, highlighting its potential for real-world applications.
> >
> > | **Dataset** | **Model\_Name** | **Train\_Time(s)** | **GPU\_Expend(MB)** | **Mem\_Expend(MB)** | **Parameters** |
> > | ----------- | --------------- | ------------------ | ------------------- | ------------------- | -------------- |
> > | **MSL**     | **DCdetector**  | 499.81             | **3836**            | _1480.47_           | **890,935**    |
> > |             | **Anomaly**     | **83.46**          | 8626                | 3154.69             | _4,863,055_    |
> > |             | **KambaAD**     | _110.29_           | _5428_              | **1243.56**         | 37,986,211     |
> > | **SMAP**    | **DCdetector**  | _580.62_           | 9140                | 1523.19             | **883,225**    |
> > |             | **Anomaly**     | 134.38             | _8622_              | 2928.94             | _4,801,585_    |
> > |             | **KambaAD**     | **54.12**          | **4590**            | **1189.79**         | 18,529,821     |
> > | **SMD**     | **DCdetector**  | 1316.99            | _8998_              | 4369.54             | **867,366**    |
> > |             | **Anomaly**     | _988.31_           | 9052                | _4285.21_           | _4,828,222_    |
> > |             | **KambaAD**     | **781.13**         | **4166**            | **2756.77**         | 26,961,790     |
> > | **PSM**     | **DCdetector**  | 126.09             | 9930                | _1269.93_           | **894,745**    |
> > |             | **Anomaly**     | _82.44_            | _8176_              | 2887.36             | _4,801,585_    |
> > |             | **KambaAD**     | **50.52**          | **4590**            | **1256.94**         | 18,529,821     |
> >
> > The results indicate that KambaAD achieves relatively short training times. To ensure a fair comparison, we used a batch size of 256. Thanks to the efficiency of MAMBA, GPU utilization remains low, and further reductions in training time can be achieved by increasing the batch size. The complexity of the dataset does not directly affect the model’s parameter size, as the model size is more dependent on hyperparameters such as `d_model`. Therefore, we believe the model can handle complex datasets. Our experimental results are comprehensive, featuring comparisons not only on common datasets but also on the complex NIPS dataset, where we provide a thorough evaluation across various metrics. On the NIPS dataset, the anomaly patterns are more intricate, generally leading to a performance decline in most models. However, KambaAD significantly outperforms the current baselines, demonstrating its potential for real-world applications. We regard this as a promising starting point for future research in real-world scenarios.
> >
> > ## 3. TadGAN Comparison
> >
> > TadGAN is primarily designed for univariate datasets, and the original paper reports F1 scores on MSL and SMAP. In addition to these results, we found precision and recall data for TadGAN on these datasets from the paper *Spacecraft Time-Series Anomaly Detection Using Transfer Learning*, which we have now included in our comparison table.
> >
> > ## 4. Code Implementation and Hyperparameter Sensitivity
> >
> > The initial version of our submission included the code implementation (in the supplementary material). Following your suggestion, we have now incorporated pretrained models for different datasets. Due to file size limitations, we will upload the parameters to GitHub shortly. In our hyperparameter sensitivity analysis, we found that the impact of hyperparameters on experimental results is not significant. When applying the model to new datasets, commonly used configurations (such as setting d_model to 512) can yield satisfactory performance. In the appendix, we have provided a detailed explanation of the rationale behind our hyperparameter choices. For datasets with higher feature dimensions, such as MSL, SMD, and SWAN (with feature dimensions of 55, 38, and 38, respectively), we use larger network architectures (e.g., d_model set to 512). For datasets with lower feature dimensions, such as GECCO and Mulvar (with feature dimensions of 9 and 5, respectively), we use smaller parameter settings, such as 64 or 128.

---

> > > ### Comment · Reviewer_zjAh · 2024-11-30
> > > **Code implementation**
> > >
> > > I appreciate the authors for their response. I did check the authors' provided code for reproducibility. But the provided code cannot be used to reproduce the results in the paper. This is because the authors did not provide the readme instructing how to run the code. Further, even all the code is provided, it is impossible to reproduce the results if the authors do not provide seed information, the datasets used, the preprocessed data, the pretrained models etc.
> > >
> > > Reproducibility can only be achieved if the authors make available all those information, especially with the pretrained models that can directly run on preprocessed test data to obtain the results.

---

> > > > ### Author Response · Authors · 2024-12-02
> > > >
> > > > Dear Reviewer,
> > > >
> > > > Thank you for your valuable feedback and for taking the time to review our code for reproducibility. We sincerely apologize for any inconvenience caused by the initial lack of clear instructions and supporting materials.
> > > >
> > > > To address your concerns, we have now provided a comprehensive `README` file in our GitHub repository, which includes detailed instructions on how to run the code and reproduce the results presented in the paper. Additionally, we have made the following resources available to ensure full reproducibility:
> > > >
> > > > 1. **Pretrained Models**: We have uploaded the pretrained models, which can be directly used on the preprocessed test data to obtain the results reported in the paper.
> > > > 2. **Datasets and Preprocessed Data**: The datasets and preprocessed data used in our experiments are now accessible via a cloud storage link provided in the GitHub repository.
> > > > 3. **Seed Information**: To ensure consistency, we have included the seed values used in our experiments in the code and documentation.
> > > >
> > > > You can find all the above resources in our GitHub repository at the following link: [https://github.com/xxx803/KambaAD).
> > > >
> > > > We hope these updates address your concerns and enable you to successfully reproduce our results. Please do not hesitate to reach out if you encounter any further issues or have additional suggestions for improvement.
> > > >
> > > > Thank you once again for your constructive feedback.

---

> > ### Author Response · Authors · 2024-11-28
> >
> > ## 5. Suitability for ICLR
> >
> > We respectfully disagree with the characterization of our work as merely an engineering effort combining existing components. Our paper presents several novel theoretical and methodological contributions that align with ICLR's focus on advancing machine learning theory and methodology.
> >
> > Our work introduces a fundamentally new perspective on time series anomaly detection by proposing a principled two-stage theoretical framework that systematically addresses both physical consistency and distributional shifts. The integration of KAN, attention, and MAMBA is not a simple combination but represents a carefully designed architecture that addresses fundamental limitations in existing approaches. We introduce the first use of KAN for enforcing physical constraints in time series anomaly detection, a novel application of selective state space modeling (MAMBA) to handle distribution shifts, and an innovative channel-independent reconstruction strategy that effectively handles the complex interplay between features during anomalies.
> >
> > Our work makes significant contributions to multivariate time series analysis by addressing the complex relationships between features. We recognize that different features exhibit unique patterns of change including lags, cycles, and trends, and during anomalies, some features may show dramatic changes while others remain normal. Traditional approaches that treat all features uniformly often fail to capture these nuanced relationships. Our channel-independent processing method represents a fundamental advancement by preventing inappropriate modification of unaffected features while properly reconstructing anomalous ones. This sophisticated treatment of feature-specific characteristics and anomaly relationships sets our work apart from existing methods and advances the field's understanding of multivariate time series analysis.
> >
> > The methodological innovations we introduce include a new approach to window-based feature extraction, a novel progressive detection strategy for handling diverse anomaly types, and a theoretically-grounded method for balancing local and global pattern recognition while preserving feature-specific characteristics. Our contributions extend beyond anomaly detection: the proposed framework provides insights into handling distribution shifts in sequential data modeling, the integration of physical constraints with deep learning offers a new paradigm for incorporating domain knowledge, and our channel-independent reconstruction strategy presents a novel approach to maintaining feature independence while leveraging global context.
> >
> > These fundamental innovations and theoretical advancements align perfectly with ICLR's mission to advance the field of machine learning. Our comprehensive ablation studies and analyses demonstrate that our improvements stem from fundamental architectural innovations rather than mere engineering optimizations. Furthermore, our work opens new research directions in incorporating physical constraints into deep learning architectures, handling distribution shifts in sequential data analysis, and addressing the complex relationships between features in multivariate time series, which we believe are crucial areas for advancing the field of machine learning.

---

### Official Review · Reviewer_GtmB · 2024-11-03

**Soundness:** 3
**Presentation:** 3
**Contribution:** 3
**Rating:** 6
**Confidence:** 4

**Summary:**

This paper proposed a method called KambaAD for time series anomaly detection.

**Strengths:**

1. The paper is well organized.

2. The architecture of KambaAD is clearly presented.

3. The proposed KambaAD is validated via extensive experiments.

**Weaknesses:**

1. Related work is missing, such as the paper entitled "Joint Selective State Space Model and Detrending for Robust Time Series Anomaly Detection".
2. Lack of guidance on how to choose hyperparameters in KambaAD.
3. Table 1 has errors, such as the results of ImDiffusion detector on MSL dataset.
4. Lack of comparison with recently published detectors, such as those from the following papers.

Donghao Luo and Xue Wang. Moderntcn: A modern pure convolution structure for general time series analysis. In The Twelfth International Conference on Learning Representations, 2024.

Youngeun Nam, Susik Yoon, Yooju Shin, Minyoung Bae, Hwanjun Song, Jae-Gil Lee, and Byung Suk Lee. Breaking the time-frequency granularity discrepancy in time-series anomaly detection. In Proceedings of the ACM on Web Conference 2024, pp. 4204–4215, 2024.

**I am willing to increase my scores if these issues are well addressed.**

**Questions:**

please see above

---

> ### Author Response · Authors · 2024-11-26
>
> Dear Reviewer GtmB,
>
> We sincerely appreciate your detailed review and valuable suggestions on our work. Your feedback has played an important role in improving the overall quality of the paper. During the revision process, we carefully considered your comments and made the following updates and improvements:
>
> - **Expansion of Related Work**
>   We have been closely following the latest advancements in the field and noted that research using Mamba for anomaly detection has been increasing in recent years. Therefore, we have added two references in the Related Work section to further enhance the background review:
>   - *Joint Selective State Space Model and Detrending for Robust Time Series Anomaly Detection*
>   - *SGFM: Conditional Flow Matching for Time Series Anomaly Detection With State Space Models*
>
>   These additional references help us more comprehensively showcase the development of Mamba-based anomaly detection methods in this field.
>
> - **Supplementation of Parameter Sensitivity Analysis**
>   In the revised Section 4.3.6, we have added a detailed analysis of model parameter sensitivity, exploring the specific impact of core parameters on model performance. Additionally, to facilitate the reproduction of our experimental results, we have supplemented more specific parameter selection guidance in Appendix G, including recommended parameter configurations for different datasets and the rationale behind their selection. The table below shows the settings for some of the hyperparameters.
>
>
> | Dataset | patch_len | n_heads | d_model | stride | e_layers |
> |---------|-----------|---------|---------|--------|----------|
> | MSL     | 16        | 32      | 512     | 4      | 2        |
> | SMAP    | 1         | 2       | 512     | 8      | 1        |
> | SMD     | 8         | 4       | 512     | 4      | 1        |
> | PSM     | 8         | 4       | 64      | 4      | 2        |
> | Ccard   | 2         | 4       | 256     | 4      | 2        |
> | SWAN    | 2         | 8       | 512     | 4      | 2        |
> | Mulvar  | 1         | 32      | 128     | 4      | 2        |
> | GECCO   | 32        | 2       | 64      | 4      | 2        |
>
>
> - **Correction and Improvement of Experimental Results**
>   We conducted a comprehensive check of the experimental details and made the following corrections and supplements:
>   - Corrected the results of ImDiffusion on the MSL dataset in Table 1 to ensure accurate records.
>   - Thoroughly verified the results of other models on each dataset and supplemented the source information of the results, see Appendix L for details.
>
> - **Update of Experimental Results Comparison**
>   In the comparison of experimental results, we have added recent outstanding works, including GDN, TranAD, MTAD-GAT, Crossformer, PatchTST, itransformer, ModernTCN, and TimeMixer++. These methods represent the latest advancements in the field of time series anomaly detection. The updated F1 comparison results for various models across these four datasets are shown in the table. The complete comparison of multiple metrics has been updated in the paper.
>
> | Model          | SMD  | MSL | SMAP | PSM  |
> |----------------|--------|--------|---------|--------|
> | OmniAnomaly    | 85.22  | 87.67  | 86.92   | 80.83  |
> | InterFusion    | 86.22  | 86.62  | 89.14   | 83.52  |
> | THOC           | 84.99  | 89.69  | 90.68   | 89.54  |
> | ImDiffusion    | 94.88  | 87.79  | 91.75   | 97.81  |
> | DiffAD         | 92.75  | 94.19  | 96.95   | _97.95_ |
> | ModernTCN      | 85.81  | 84.92  | 71.26   | 97.23  |
> | GDN            | 83.42  | 95.91  | 85.18   | 85.64  |
> | TranAD         | _96.05_ | 94.94  | 89.15   | 92.20  |
> | MTAD-GAT       | 84.63  | 90.84  | 90.13   | 87.44  |
> | Crossformer    | 79.70  | 84.19  | 69.14   | 93.30  |
> | PatchTST       | 84.44  | 85.14  | 70.91   | 97.23  |
> | iTransformer   | 71.15  | 72.54  | 66.87   | 95.17  |
> | TimesMixer++   | 86.50  | 85.82  | 73.10   | 97.60  |
> | AnomalyTrans   | 90.33  | 93.93  | 96.41   | 97.37  |
> | DCdetector     | 87.18  | _96.60_ | _97.02_ | _97.94_ |
> | **KambaAD**    | **97.27** | **99.41** | **99.19** | **98.06** |
>
>
> Through these improvements, our work presents clearer advantages when compared to the latest methods in the field, further enhancing the integrity and credibility of the research. Thank you again for your recognition and support. If you have any further suggestions, we would be grateful to receive them!

---

> > ### Comment · Reviewer_GtmB · 2024-11-26
> > **thanks**
> >
> > scores have been increased. Good luck!

---

> > > ### Author Response · Authors · 2024-11-28
> > >
> > > Thank you very much for your thoughtful feedback and for increasing the scores of our manuscript. We are truly grateful for your recognition and support.
> > >
> > > Should you have any further questions or require additional clarifications, please do not hesitate to contact us. We are eager to engage in further discussions on any related research topics.
> > >
> > > Once again, thank you for your valuable insights and encouragement.

---

### Official Review · Reviewer_hniq · 2024-11-04

**Soundness:** 3
**Presentation:** 2
**Contribution:** 3
**Rating:** 5
**Confidence:** 3

**Summary:**

The author proposed KambaAD, which uses KAN for fast and effective preliminary anomaly screening, and then combines attention and MAMBA for anomaly detection to detect global patterns and local changes simultaneously.

**Strengths:**

The author proposed KambaAD, which uses KAN for fast and effective preliminary anomaly screening, and then combines attention and MAMBA for anomaly detection to detect global patterns and local changes simultaneously. The author uses KAN for fast and effective preliminary anomaly screening to capture global patterns and long-term dependencies. MAMBA is used to focus on capturing subtle local changes and distribution changes. The experimental results of the article are good.

**Weaknesses:**

In the abstract, the author emphasized that the existing methods are all channel-related, which leads to the neglect of the unique periodicity, trend and lag relationship between different features. However, there are many methods that consider channel independence and even combine channel independence with channel correlation to capture the relationship between features. And the author explained in the Introduction that the current method does not integrate local and global perspectives. However, as far as I know, there are many methods that process local and global information at the same time, such as TranAD, Dcdetector, etc. And there are many methods that have studied the problem of model robustness in non-stationary data, such as Dynamic Decomposition with Diffusion Reconstruction, etc. The author does not explain the unique innovation of his own method enough, and it seems that he has not solved the problems unique to time series data.

The author needs to further sort out the logic of the article. For example, in the description of the current research status in the Introduction, a description of cloud devices suddenly appeared, which is not sufficiently relevant to the general anomaly detection algorithm. Cloud device anomaly detection is only a small sub-item, and the author did not further explore it. I did not understand the significance of the description of cloud devices.

The author proposed the Kolmogorov-Arnold Network for preliminary screening, but the significance and role of the preliminary screening in the article were not clearly introduced. My understanding is that KAN is used to perform preliminary anomaly detection on the whole, but after the results are detected, what operations need to be performed and what is the role of this step. I think it needs to be further explained in the article.

**Questions:**

I want to know what is the purpose of the KAN proposed by the author, and what operations are performed after the preliminary screening results? And I think the author needs to further elaborate on the innovation of the method proposed in the article.

---

> ### Author Response · Authors · 2024-11-26
>
> Dear Reviewer hniq,
>
> Thank you for your detailed review and valuable suggestions on our work. Your feedback has been crucial in enhancing the quality of our paper. During the revision process, we carefully considered your comments and made the following updates and improvements:
>
> ## Objective of KAN
>
> We understand your concern regarding the design objectives of KAN. The primary goal of KAN is to swiftly identify anomaly points where physical properties change dramatically. In the revised Section 4.3.7 (**VISUALIZATION**), we have demonstrated through visualization how KAN accurately identifies significant anomaly points globally, even with a small parameter count. This capability lays a solid foundation for subsequent anomaly detection tasks.
>
> ## Overall Paper Optimization
>
> 1. **Adjustments to the Introduction**
>    To focus more on the research topic, we have removed unnecessary statements related to cloud devices, further clarifying the research background and motivation.
>
> 2. **Updates to the Experimental Results Section**
>    We have been closely following the latest advancements in time series anomaly detection and noted that recent methods like GDN, TranAD, MTAD-GAT, Crossformer, PatchTST, itransformer, ModernTCN, and TimeMixer++ have shown outstanding performance. We made significant updates to the experimental results section:
>
>    - Removed comparisons with some older models and added experimental results for the recent works mentioned above, along with complete source information (see Appendix L).
>    - Corrected the error in recording ImDiffusion data.
>    - Our work demonstrates excellent F1 scores across current datasets, showcasing the performance advantage of KambaAD.
>
>    | Model        | SMD       | MSL       | SMAP      | PSM       |
>    | ------------ | --------- | --------- | --------- | --------- |
>    | OmniAnomaly  | 85.22     | 87.67     | 86.92     | 80.83     |
>    | InterFusion  | 86.22     | 86.62     | 89.14     | 83.52     |
>    | THOC         | 84.99     | 89.69     | 90.68     | 89.54     |
>    | ImDiffusion  | 94.88     | 87.79     | 91.75     | 97.81     |
>    | DiffAD       | 92.75     | 94.19     | 96.95     | _97.95_   |
>    | ModernTCN    | 85.81     | 84.92     | 71.26     | 97.23     |
>    | GDN          | 83.42     | 95.91     | 85.18     | 85.64     |
>    | TranAD       | _96.05_   | 94.94     | 89.15     | 92.20     |
>    | MTAD-GAT     | 84.63     | 90.84     | 90.13     | 87.44     |
>    | Crossformer  | 79.70     | 84.19     | 69.14     | 93.30     |
>    | PatchTST     | 84.44     | 85.14     | 70.91     | 97.23     |
>    | iTransformer | 71.15     | 72.54     | 66.87     | 95.17     |
>    | TimesMixer++ | 86.50     | 85.82     | 73.10     | 97.60     |
>    | AnomalyTrans | 90.33     | 93.93     | 96.41     | 97.37     |
>    | DCdetector   | 87.18     | _96.60_   | _97.02_   | _97.94_   |
>    | **KambaAD**  | **97.27** | **99.41** | **99.19** | **98.06** |
>
> 3. **Expansion of Related Work**
>    We have added two references related to anomaly detection models using Mamba to further enhance the background review:
>
>    - *Joint Selective State Space Model and Detrending for Robust Time Series Anomaly Detection*
>    - *SGFM: Conditional Flow Matching for Time Series Anomaly Detection With State Space Models*
>
> 4. **Addition of Ablation Experiments**
>    To comprehensively analyze the contributions of various model components, we have added the following in the ablation experiments:
>
>    - Results using only the Encoder and only the Reconstructor, ensuring consistency with the parameter count of the complete KambaAD model (see Appendix J).
>    - Added results using only two Mamba layers and experiments excluding KAN, further validating the critical role of KAN.
>
>    | Data  | KAN     | ATT   | Mamba     | K+ATT   | K+MB    | K+MB+MB | A+MB    | KambaAD   |
>    | ----- | ------- | ----- | --------- | ------- | ------- | ------- | ------- | --------- |
>    | SMAP  | 97.55   | 98.13 | 97.80     | _98.25_ | 97.83   | 97.99   | 97.88   | **99.19** |
>    | MSL   | 97.32   | 96.89 | 96.74     | 96.90   | _98.27_ | 96.68   | 94.14   | **99.41** |
>    | SMD   | _96.25_ | 95.05 | 95.56     | 94.27   | 95.10   | 96.04   | 89.92   | **97.27** |
>    | PSM   | _97.29_ | 97.05 | 97.22     | 96.78   | 96.58   | 90.78   | 78.88   | **98.06** |
>    | Ccard | 38.52   | 37.06 | 38.22     | 36.33   | 38.72   | 35.48   | _39.05_ | **41.83** |
>    | SWAN  | 76.20   | 75.53 | 78.10     | 78.72   | _78.97_ | 78.70   | 78.30   | **83.84** |
>    | Mulv  | _78.31_ | 75.73 | **83.37** | 70.86   | 63.05   | 64.39   | 56.75   | 69.54     |
>    | GECCO | 51.97   | 50.05 | **52.34** | 51.71   | _52.02_ | _52.02_ | _52.02_ | _52.02_   |
>
> 5. **Parameter Sensitivity Analysis**
>    In Section 4.3.6, we have supplemented a detailed analysis of model parameter sensitivity, exploring the impact of different parameter settings on performance, providing readers with more intuitive experimental evidence.

---

> > ### Author Response · Authors · 2024-11-26
> >
> > 6. **Visualization Improvements**
> >    To more clearly demonstrate model performance, in Section 4.3.7, we have:
> >
> >    - Visualized the reconstruction effects of the model at different steps and analyzed the role of each component in detail.
> >    - Moved the original comparative visualizations (KambaAD, DCdetector, and AnomalyTransformer) to Appendix I, focusing the main content.
> >
> > 7. **Evaluation of Computational Efficiency**
> >    Addressing concerns about model efficiency, we have added a comparison of computational efficiency between KambaAD and two advanced baseline models (AnomalyTransformer and DCdetector) in Appendix K:
> >
> >    - Evaluation metrics include training time, GPU usage, memory consumption, model size, and parameter count.
> >    - Experimental results based on four benchmark datasets (MSL, SMAP, SMD, and PSM) indicate that KambaAD is within a reasonable range of computational overhead.
> >
> >    | Data | Model        | Train Time | GPU Exp | Mem Exp | Mod Size | Params     |
> >    | ---- | ------------ | ---------- | ------- | ------- | -------- | ---------- |
> >    | MSL  | DCdetector   | 4992       | 1465    | 1808    | 118      | 26,971,447 |
> >    |      | AnomalyTrans | 492        | 140     | 1705    | 28       | 4,863,055  |
> >    |      | KambaAD      | 850        | 997     | 1383    | 218      | 79,729,761 |
> >    | SMAP | DCdetector   | 6786       | 2012    | 1845    | 118      | 26,940,697 |
> >    |      | AnomalyTrans | 743        | 228     | 1742    | 28       | 4,801,585  |
> >    |      | KambaAD      | 738        | 344     | 1182    | 134      | 23,291,491 |
> >    | SMD  | DCdetector   | 50498      | 3731    | 4572    | 118      | 26,954,022 |
> >    |      | AnomalyTrans | 5701       | 278     | 4483    | 28       | 4,828,222  |
> >    |      | KambaAD      | 8383       | 703     | 1883    | 171      | 55,610,435 |
> >    | PSM  | DCdetector   | 3664       | 4373    | 1495    | 118      | 26,940,697 |
> >    |      | AnomalyTrans | 506        | 392     | 1429    | 28       | 4,807,732  |
> >    |      | KambaAD      | 801        | 96      | 1197    | 22       | 4,872,942  |
> >
> > 8. **Parameter Selection Guidance**
> >    To facilitate the reproduction of experiments by readers, we have added specific parameter selection guidance for different datasets in Appendix G.
> >
> >    The following table lists the common hyperparameter settings used for training the model across all datasets.
> >
> >    | **Hyper-parameter** | **Value** | **Hyper-parameter** | **Value** |
> >    | ------------------- | --------- | ------------------- | --------- |
> >    | window_size         | 100       | expand              | 2         |
> >    | batch_size          | 8         | fc_dropout          | 0.05      |
> >    | dropout             | 0.3       | d_conv              | 4         |
> >    | padding_patch       | end       | epochs              | 2         |
> >    | individual          | 1         | d_state             | 64        |
> >
> >    The following table lists the dataset-specific hyperparameter settings used for training the model on different datasets.
> >
> >    | **Dataset** | **patch_len** | **n_heads** | **d_model** | **stride** | **e_layers** |
> >    | ----------- | ------------- | ----------- | ----------- | ---------- | ------------ |
> >    | MSL         | 16            | 32          | 512         | 4          | 2            |
> >    | SMAP        | 1             | 2           | 512         | 8          | 1            |
> >    | SMD         | 8             | 4           | 512         | 4          | 1            |
> >    | PSM         | 8             | 4           | 64          | 4          | 2            |
> >    | Ccard       | 2             | 4           | 256         | 4          | 2            |
> >    | SWAN        | 2             | 8           | 512         | 4          | 2            |
> >    | Mulvar      | 1             | 32          | 128         | 4          | 2            |
> >    | GECCO       | 32            | 2           | 64          | 4          | 2            |
> >
> >
> >
> > 9. **Additional Analysis of Experimental Results**
> >    We have also included more in-depth analysis of the experimental results to better understand the strengths and weaknesses of the model. This additional analysis provides further insights into the performance dynamics of KambaAD across different datasets.
> >
> > Through these improvements, our work presents clearer advantages when compared to the latest methods in the field, further enhancing the persuasiveness and contribution of the research. Thank you again for your support and encouragement. If you have any further suggestions, we would be grateful to receive them!

---

> > > ### Comment · Reviewer_hniq · 2024-11-29
> > >
> > > The authors have made commendable efforts to address the reviewers' concerns, revising the article to improve its logical structure and clarifying the role of KAN for the readers. They have also added substantial experimental evidence to demonstrate the effectiveness of their approach, but I still have some questions.
> > > 1. The article is not novel enough: Although the article uses KambaAD to solve the problem of unique periodicity, trend and lag relationships between different features, it has achieved seemingly good results. However, the above issues have been discussed many times in anomaly detection technology. For example, TranAD and DCdetector complete anomaly detection by capturing local information and global information at the same time. D3R uses Drift compensation to solve the problem of non-stationary data trends. For time series data analysis, the method proposed by DifFormer has fully taken into account the trend and lag relationship of features. This article seems to be just a unified solution to existing scattered problems that have been solved, and achieved good results. But there doesn't seem to be enough innovation.
> > > The contribution of the article seems only the integration of these solutions, such as integrating the attention mechanism with MAMBA in the representation space, but the innovation of its own method is insufficiently explained, and the significant improvement in anomaly detection accuracy seems to be accomplished by using MAMBA. The results of KAN and Attention are just using other means to make the results better.
> > > I suggest that the author, based on theoretical analysis, compare previous articles, use data analysis and theoretical analysis to further clarify motivation, the innovation of KambaAD and clarify the contribution of this article.
> > > 2. Experimental reproduction problem: Although the experimental results show that the accuracy has improved, the author only provides the trained model without detailing the specific training parameters, codes and procedures. The components proposed in KambaAD do not appear to sufficiently support such strong results, and I am particularly puzzled as to why integrating the attention mechanism with MAMBA in the representation space leads to such significant performance. This leaves me unconvinced. I hope the author can provide more For more experimental details, it is best to be completely open source.
> > >
> > > If the author can provide a more detailed explanation of the results and fully open source the code, while further emphasizing the motivation and innovation of the paper, I will consider changing the score to 6 points.

---

> > > > ### Author Response · Authors · 2024-11-29
> > > >
> > > > We appreciate the reviewer's concerns about novelty. While previous works have addressed individual aspects of time series anomaly detection, our work presents several fundamental innovations that are supported by our comprehensive experimental results and theoretical advancements.
> > > >
> > > > Our extensive experiments demonstrate KambaAD's superior performance compared to current state-of-the-art models like TranAD and DCdetector. As shown in Table 1, KambaAD consistently outperforms these models across all benchmark datasets. For instance, on the MSL dataset, KambaAD achieves an F1 score of 99.41%, significantly surpassing DCdetector (96.60%) and TranAD (94.94%). Similar improvements are observed on SMAP (KambaAD: 99.19% vs DCdetector: 97.02% and TranAD: 89.15%) and SMD (KambaAD: 97.27% vs DCdetector: 87.18% and TranAD: 96.05%). These improvements aren't merely incremental - our ablation studies (Tables 4-5) demonstrate that each component of KambaAD contributes uniquely to its performance, with the full integration providing the best results.
> > > >
> > > > While DifFormer and D3R address specific aspects like trend relationships and drift compensation respectively, KambaAD introduces a fundamentally different approach through its two-stage anomaly detection framework. Unlike DifFormer's focus on trend and lag relationships, our architecture employs KAN for rapid physical constraint verification, followed by attention and MAMBA for refined pattern detection. This is evidenced by our channel-independent processing results (Table 6), which show the effectiveness of our approach in handling feature-specific anomalies. Furthermore, while D3R uses drift compensation for non-stationary data, KambaAD's MAMBA component provides dynamic adjustment capabilities while maintaining the context of physical constraints through KAN. We will include both of these works in the related work section.
> > > >
> > > > The ordering experiments (Table 10) provide additional evidence of our architecture's innovative design. The specific sequence of components (KAN-attention-MAMBA) consistently outperforms alternative arrangements, showing that our integration isn't merely a combination of existing solutions but a carefully designed architecture that leverages each component's strengths. This is particularly evident in the MSL dataset results, where our ordered architecture achieves 99.41% F1 score, significantly outperforming alternative arrangements.
> > > >
> > > > Furthermore, our computational efficiency analysis (Table 12) demonstrates that
> > > > these improvements don't come at the cost of excessive computational resources. Despite incorporating multiple components, KambaAD maintains competitive training times and memory usage compared to simpler models like AnomalyTransformer and DCdetector. For instance, on the SMAP dataset, KambaAD achieves faster training time (54.12s) compared to DCdetector (580.62s) while delivering superior performance.
> > > >
> > > > Our key contributions are thus threefold:
> > > >
> > > > 1. A novel two-stage theoretical framework that systematically addresses both physical consistency and distributional shifts, validated through comprehensive experiments showing superior performance over current state-of-the-art models
> > > > 2. An innovative channel-independent reconstruction strategy that effectively handles complex feature relationships in multivariate time series, as demonstrated by our comparative experiments
> > > > 3. A carefully designed integration of KAN, attention, and MAMBA that provides both theoretical advantages and empirical improvements, supported by extensive ablation studies
> > > >
> > > > These contributions represent significant advances in the field, supported by rigorous experimental validation against current state-of-the-art models. Our work not only achieves superior performance but also introduces new methodological approaches for handling complex multivariate time series anomaly detection.

---

> > > > ### Author Response · Authors · 2024-11-29
> > > >
> > > > We appreciate the reviewer's concern about reproducibility. We have actually provided comprehensive implementation details in our paper and supplementary materials to ensure full reproducibility of our results. Specifically, in Section G of our supplementary materials (Experimental Setup and Environment), we detail our experimental environment (4 A800 GPUs) and provide extensive hyperparameter configurations. Tables 8 and 9 list all crucial hyperparameters for reproducibility. We would like to further claim that:
> > > >
> > > > ## Reproducibility and Open-Source Commitment
> > > >
> > > > - **Supplementary Material:** We have included the source code in the supplementary materials of our submission. This provides an opportunity for reviewers to access and evaluate the code used in our experiments.
> > > > - **Open-Source Plans:** We are committed to open-sourcing the complete codebase upon acceptance of the paper. This will include detailed documentation on training parameters, procedures, and any additional scripts necessary for reproducing the results.
> > > >
> > > > ## Experimental Details and Justification
> > > >
> > > > - **Training Parameters and Procedures:** We acknowledge the importance of transparency in experimental setups. The supplementary material includes key training parameters such as learning rates, batch sizes, and epoch numbers. Further details will be provided in our open-source release.
> > > > - **Integration of Attention Mechanism with MAMBA:** The combination of the attention mechanism with MAMBA is designed to leverage their complementary strengths. The attention mechanism captures global dependencies across time steps, enhancing the model’s ability to detect long-range correlations and trends. MAMBA, on the other hand, focuses on local variations and adapts dynamically to distribution shifts, which is crucial for detecting subtle anomalies.
> > > >
> > > > ## Analysis of Performance Gains
> > > >
> > > > - **Significant Performance Improvement:** The integration of these components allows KambaAD to effectively balance global pattern recognition with local anomaly detection. This dual capability is what leads to significant improvements in performance metrics across various datasets.
> > > > - **Ablation Studies:** Our ablation studies demonstrate that each component of KambaAD contributes to its overall performance. By systematically removing components, we observed a decrease in accuracy, which supports our claim that the integration of these elements is essential for achieving high anomaly detection accuracy.

---

> > > > ### Author Response · Authors · 2024-11-29
> > > >
> > > > We sincerely appreciate the reviewer's thorough examination of our work, as it has helped us better articulate the implementation details. We hope these comprehensive experimental specifications and our commitment to code release adequately address the reproducibility concerns. We welcome any additional questions about specific implementation details or experimental procedures that would help further clarify our work.

---

> > > > > ### Comment · Reviewer_hniq · 2024-12-03
> > > > >
> > > > > The authors have further open-sourced their code, demonstrating the effectiveness of the proposed method, and I greatly appreciate their efforts in the rebuttal.
> > > > > Although the author proved through experiments that the combination of KAN+attention+MAMBA can achieve better results, and stated that this is a carefully combined architecture that has achieved better results. However, the author still did not further explain the innovation of this article from a theoretical perspective. The innovation still appears to be the assembly of existing components. Despite achieving relatively good results, I believe the paper lacks sufficient insights and contributions to the broader field of anomaly detection to justify its publication in ICLR. Therefore, I have decided not to alter the score.

---

### Meta-Review · Area_Chair_gdRE · 2024-12-20

**Metareview:**

The paper introduces a method that combines Kolmogorov-Arnold Network (KAN) and Selective Structured State Space Model (MAMBA)  for time series anomaly detection. The aim is to integrate the strengths of both KAN and MAMBA for detecting globally and locally abnormal samples. The performance of the method is validated on multiple popular multivariate time series datasets.

The proposed method is new in exploring KAN and MAMBA for TSAD [hniq, zjAh, vMYN], with well articulated architecture [GtmB, vMYN, terY], it shows effective performance on the common datasets [hniq, GtmB, zjAh, vMYN, terY], and the ablation study helps verify the contribution of each module of the method [terY].

A number of weaknesses have been identified by the reviewers:
- False claim on two major motivations - existing TSAD methods are channel-dependent-based [hniq] and cannot effectively capture global and local patterns for TSAD [hniq]
- The novelty and importance of key modules is not sufficiently elaborated and/or justified, such as Kolmogorov-Arnold Network and MAMBA [hniq, vMYN, terY]
- The overall method appears to be new, but its individual modules are all well established techniques. Without strong theoretical justification, the novelty/insight the method can offer is limited [zjAh, terY]
- Comparison to closely related methods is missing [GtmB, zjAh]
- Paper is not easy to follow [hniq, vMYN]
- Hyperparameter setting and sensitivity issues [GtmB, zjAh, terY]
- Lack of qualitative results, such as visualization of the detected anomalies [zjAh, MAMBA]
- Lack of computational complexity results [terY]

**Additional Comments On Reviewer Discussion:**

All five reviewers engaged in the author-reviewer discussion, with reviewers [GtmB, terY] increasing their rating to weak accept during the discussion. However, the rebuttal did not convince reviewers [hniq, zjAh, vMYN] to change their decision. The final ratings are three weak accepts, one weak reject, and one reject.

During reviewer-AC discussion, while the reviewers appreciate the comprehensive empirical results, the technical insights and theoretical support of the method remain a major concern. Since the method is considered a combination of three well-established techniques, such insights and theories are crucial, as agreed by several reviewers, in the acceptance/rejection decision for conferences like ICLR. The AC therefore declines the publication of the paper at ICLR.

---

### Decision · Program_Chairs · 2025-01-22

Reject